# Agonist immunotherapy restores T cell function following MEK inhibition improving efficacy in breast cancer

Sathana Dushyanthen[1], Zhi Ling Teo[1,2], Franco Caramia[1], Peter Savas[1], Christopher P. Mintoff[1], Balaji Virassamy[1], Melissa A. Henderson[1], Stephen J. Luen[1], Mariam Mansour[1], Michael H. Kershaw[1,2], Joseph A. Trapani[1,2], Paul J. Neeson[1], Roberto Salgado[3], Grant A. McArthur[1], Justin M. Balko[4], Paul A. Beavis[1,2], Phillip K. Darcy[1,2] & Sherene Loi[1,2]

The presence of tumor-infiltrating lymphocytes in triple-negative breast cancers is correlated with improved outcomes. Ras/MAPK pathway activation is associated with significantly lower levels of tumor-infiltrating lymphocytes in triple-negative breast cancers and while MEK inhibition can promote recruitment of tumor-infiltrating lymphocytes to the tumor, here we show that MEK inhibition adversely affects early onset T-cell effector function. We show that α-4-1BB and α-OX-40 T-cell agonist antibodies can rescue the adverse effects of MEK inhibition on T cells in both mouse and human T cells, which results in augmented anti-tumor effects in vivo. This effect is dependent upon increased downstream p38/JNK pathway activation. Taken together, our data suggest that although Ras/MAPK pathway inhibition can increase tumor immunogenicity, the negative impact on T-cell activity is functionally important. This undesirable impact is effectively prevented by combination with T-cell immune agonist immunotherapies resulting in superior therapeutic efficacy.

[1] Peter MacCallum Cancer Centre, Melbourne, VIC 3000, Australia. [2] Sir Peter MacCallum Department of Oncology, University of Melbourne, Parkville, VIC 3010, Australia. [3] Breast Cancer Translational Research Laboratory, Institute Jules Bordet, Brussels 1000, Belgium. [4] Breast Cancer Research Program and Department of Medicine, Vanderbilt-Ingram Cancer Centre and Vanderbilt University Medical Centre, Nashville, TN 37232, USA. Paul A Beavis, Phillip K Darcy, and Sherene Loi contributed equally to this work. Correspondence and requests for materials should be addressed to P.A.B. (email: Paul.Beavis@petermac.org) or to P.K.D. (email: phil.darcy@petermac.org) or to S.L. (email: sherene.loi@petermac.org)

The predictive and prognostic significance of tumor-infiltrating lymphocytes (TILs) has been highlighted in various solid cancers such as melanoma[1, 2], lung cancer[3, 4], and colorectal cancer[5, 6]. These findings suggest an important role of T-cell mediated immunosurveillance in influencing the biology of these cancers[7]. Recent research has also demonstrated the prognostic value of TILs in certain breast cancer (BC) subtypes such as HER2-positive (HER2+)[8–10] and in particular, triple-negative breast cancer (TNBC)[7, 11, 12], where the presence of higher levels of TILs in primary tumors was found to correlate with better disease free and overall survival[11–14]. These associations suggest that immunotherapies may be effective in TNBC, a BC subtype where novel therapies are urgently needed. Despite evidence for the biological importance of TILs in TNBC, mechanisms underlying heterogeneity in TIL recruitment within breast tumors remain largely unknown. Better understanding of these mechanisms will inform development of immunotherapy approaches that may favorably alter the tumor microenvironment and ultimately improve patient outcomes.

We have previously shown that oncogenic activation of the Ras/MAPK pathway is associated with significantly decreased levels of TILs and poorer survival in TNBC patients[15–18]. This

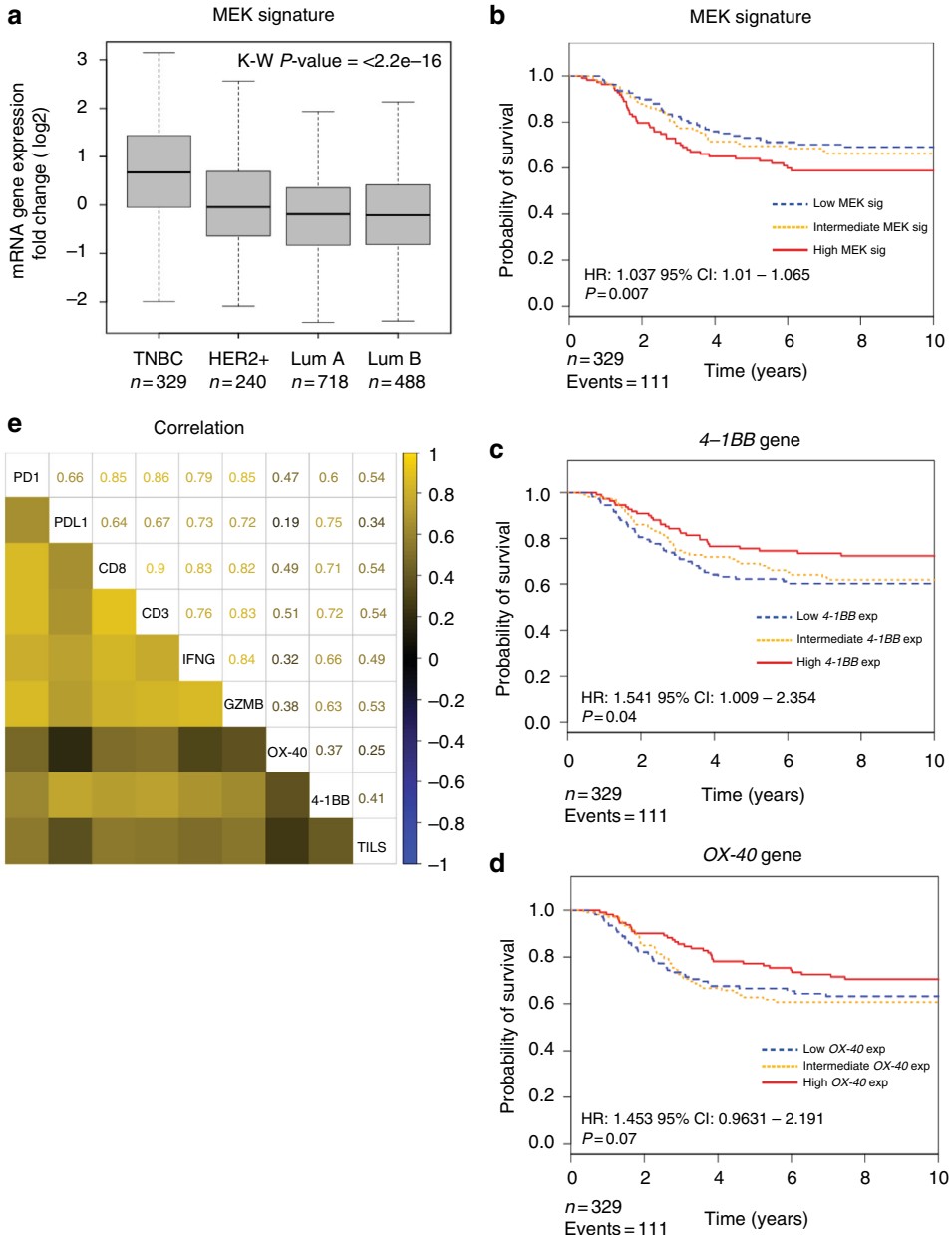

**Fig. 1** Clinical correlates of a MEK activation gene signature and 4-1BB and OX-40 gene expression in human TNBC. **a** Higher levels of the MEK gene signature in TNBC (n = 329, Kruskal–Wallis; P = 2.2e-16) compared with other breast cancer subtypes from the METABRIC data set. *Box plot* represents the lower, median and upper quartile, while *whiskers* represent the highest and lowest range for the upper and lower quartiles. P-value represents Kruskal–Wallis test **b**–**d** Kaplan–Meier survival curves of TNBC patients according to tertiles of the **b** MEK signature (HR: 1.037, 95% CI: 1.01–1.065; P = 0.007), **c** 4-1BB gene (HR: 1.541 95% CI: 1.009–2.354; P = 0.04), and **d** OX-40 gene (HR: 1.453 95% CI: 0.9631–2.191; P = 0.07), stratified by low, intermediate, and high gene expression tertiles. P-values represent Cox regression analysis. **e** Gene correlations between the TILs signature, OX-40, 4-1BB, and key breast cancer-related immune prognostic genes from the TCGA database. *Yellow* high correlation, *blue* low correlation. P-values represent Pearson's correlation coefficient

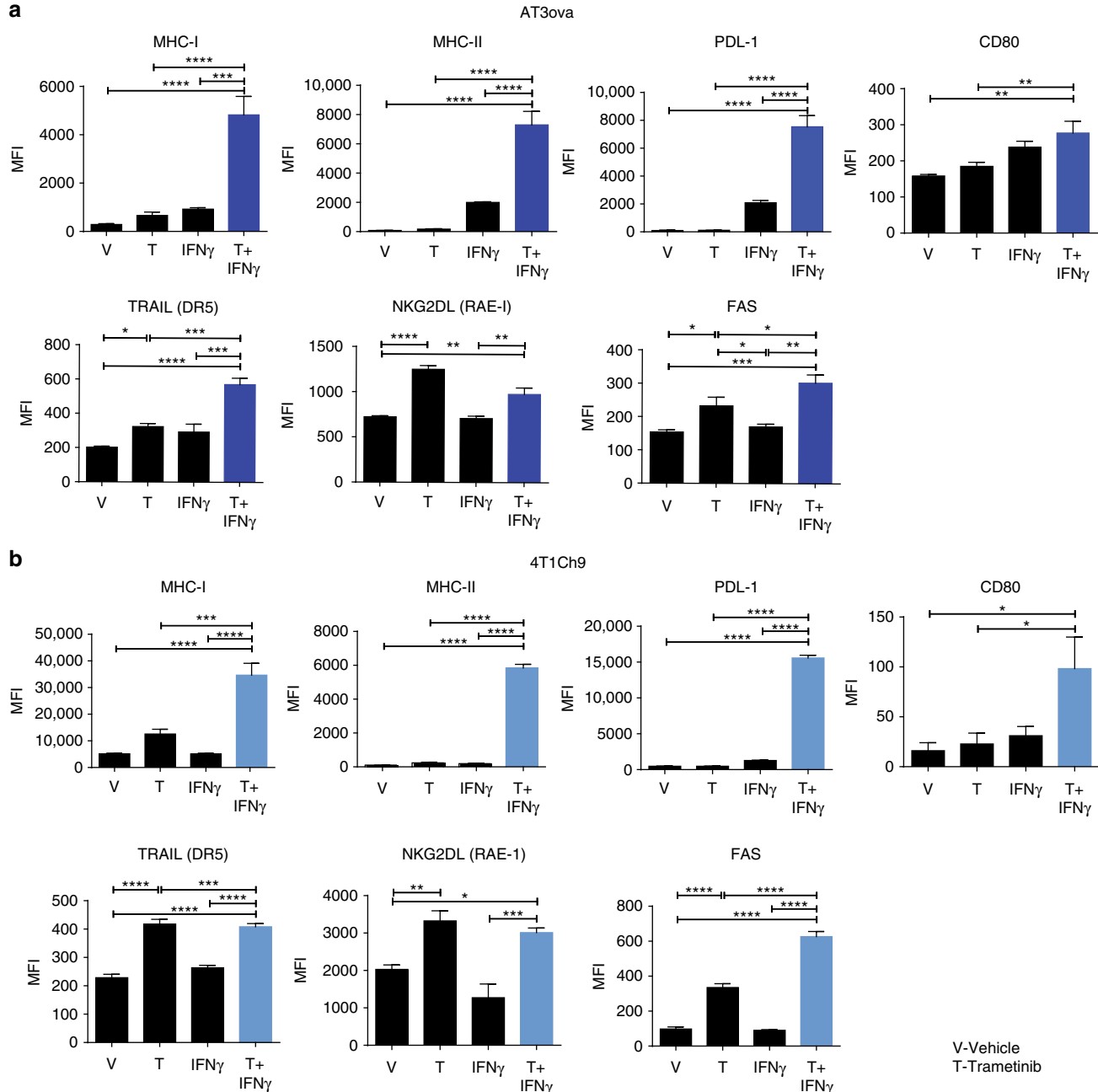

**Fig. 2** MEK inhibition increases tumor immunogenicity of AT3ova and 4T1Ch9 TNBC tumors in vitro. FACS analysis of MHC-I, MHC-II, PDL-1, CD80, TRAIL (DR5) NKG2DL (RAE-1), and Fas expression (*MFI*; mean fluorescence index) on the **a** AT3ova and **b** 4T1Ch9 murine TNBC cell lines following trametinib treatment alone (100 nM), IFNγ stimulation alone (5 ng/ml), or combination of trametinib treatment and IFNγ stimulation. Data are presented as mean ± SEM of triplicate samples. *P*-values represent one-way ANOVA and post hoc Fisher's LSD tests. *P < 0.05, **P < 0.01, ***P < 0.001, ****P < 0.0001

observation raises the possibility that Ras/MAPK pathway inhibition may relieve local immunosuppression, thereby enhancing TIL infiltrate and improving patient outcomes. Paradoxically, MEK signaling in lymphocytes is critical for CD8[+] and CD4[+] T-cell activation, proliferation, function, and survival[19, 20]. Therefore while inhibition of Ras/MAPK pathway can potentially enhance TIL numbers by enhancing tumor immunogenicity[15], theoretically it likely simultaneously inhibits effector T-cell function[21–25], though the clinical relevance of this is currently unclear. The complex interplay between the kinetics of MEK inhibition (MEKi) on T-cell function and its relevance to the therapeutic efficacy of MEKi in solid cancers is currently

undefined. Limited studies have undertaken in depth exploration into the effects of MEKi on T cell functionality, where most reports have been somewhat contradictory. Some studies have shown that MEKi potentiates anti-tumor immunity[23, 25], while others suggest that MEKi only transiently inhibits T-cell function[21, 22]. As such, in this study we aimed to investigate the long-term effects of MEKi on T cells.

Agonist antibodies such as α-4-1BB (CD137) and α-OX-40 (CD134) antibody have been shown to activate T cells independently of MEK1/2 signaling[26]. Hence, if MEKi is detrimental to T-cell function, combination with immune agonists may overcome this defect, which may lead to significantly improved

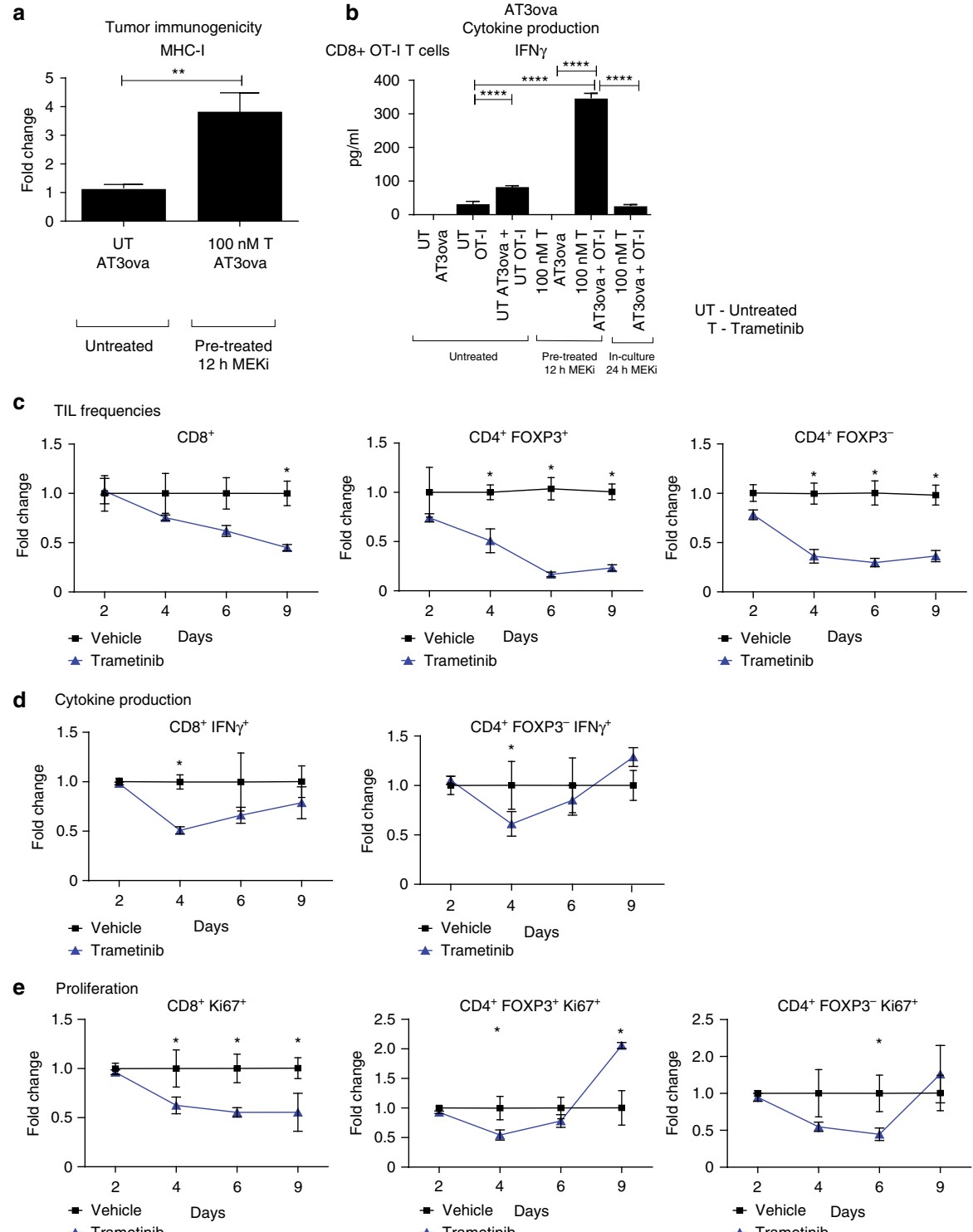

**Fig. 3** MEK inhibition increases tumor immunogenicity but reduces T-cell effector functions. Co-culture studies undertaken with AT3ova tumor cells and CD8[+] OT-I T cells; 12 h pre-treatment followed by co-culture, or 24 h co-culture with trametinib treatment. **a** FACS analysis of MHC-I expression of AT3ova tumor cells normalized to non-treated tumors. **b** IFNγ production from OT-I T cells. **c–e** Mice ($n = 3$ per group) bearing AT3ova tumors were treated with vehicle (PEG 400/solutol) or trametinib (1 mg/kg/daily), and tumors were harvested on day 2, 4, 6, and 9 post treatment. Changes in **c** TIL frequency (CD8[+], CD4[+] FOXP3[−], CD4[+] FOXP3[+]) as a proportion of CD45[+] live cells, **d** cytokine production by T cells, and **e** proliferation of T cells measured by Ki67 expression were determined ex vivo by FACS analysis. Values were normalized to vehicle controls at each time point and data are expressed as fold change for the number of positive cells. Data are presented as mean ± SEM, and is a representative of three independent repeats. *P*-values represent unpaired *t*-tests at each time point and post hoc Fisher's LSD tests. *$P < 0.05$, **$P < 0.01$, ***$P < 0.001$, ****$P < 0.0001$

therapeutic efficacy. Thus, we hypothesized that these agonists may restore effector T-cell function even in the presence of MEK1/2 inhibitors. Stimulation of these agonist pathways has been reported to lead to increased T-cell activation, proliferation, expansion, survival, memory formation, $T_H1$ development, and induction of interleukin (IL)-2 and IFNγ immune responses[27, 28]. Herein, we demonstrate that MEKi does significantly inhibit early T-cell signaling where immune agonists, α-4-1BB and α-OX-40, can effectively restore T-cell frequency, proliferation, and function. As such, our results confirm that MEKi can prime tumor immunogenicity and combination with either α-4-1BB or α-OX-40 agonist immunotherapy results in superior therapeutic efficacy due to protection of early and crucial TIL function in preclinical models of TNBC.

## Results

**MEK gene signature and prognosis in human TNBC.** Using the publicly available gene expression data of human primary TNBCs[29], we found that levels of a gene signature representing MEK activation[30] was significantly higher (Kruskal–Wallis; $P = 2.2e-16$) in TNBC (also known as "basal-like") compared with other breast cancer subtypes (Fig. 1a). Higher levels of the MEK gene signature was also significantly associated with poorer survival outcomes (Cox regression analysis, hazard ratio (HR): 1.037, 95% confidence interval (CI): 1.01–1.065; $P = 0.007$) (Fig. 1b). Conversely, higher *4-1BB* (HR: 1.541, 95% CI: 1.009–2.354; $P = 0.04$) and *OX-40* (HR: 1.453, 95% CI: 0.9631–2.191; $P = 0.07$) expression was associated with better outcomes (Fig. 1c, d). We next looked at the correlation between gene expression of 4-1BB and OX-40 with other immune genes in human breast cancers profiled in The Cancer Genome Atlas[31], where we had also evaluated TILs on the diagnostic histo-pathology slides using our previously defined method[32] (Fig. 1e). As expected, higher levels of *4-1BB* and *OX-40* were strongly correlated with increasing quantities of TILs, T-cell activation, and cytotoxic function markers, suggesting an important role of these factors in modulating a coordinated immune-mediated anti-tumor T-cell response. The strong positive correlation between TILs and 4-1BB/OX-40 expression (Fig. 1e) likely explains the association with 4-1BB/OX-40 and improved patient outcomes (Fig. 1c, d). Taken together, this data from human TNBC samples supports our rationale for evaluating Ras/MAPK targeted inhibitors (MEKi) in combination with T-cell agonist immunotherapies as a treatment strategy for TNBC.

**MEKi increases the immunogenicity of TNBCs.** We have previously shown that MEKi increases MHC-I, MHC-II, and PDL-1 on both AT3ova and 4T1Ch9 tumor cells in vivo[15]. To further characterise the effect of MEKi on tumor immunogenicity, we examined the expression of other receptors and ligands on these tumors following MEKi. We observed that the expression of Fas, TRAIL, and NKG2D (RAE-1) were significantly upregulated (one-way analysis of variance (ANOVA); $P < 0.05$) in the presence of MEKi in vitro in both the AT3ova and 4T1Ch9 cell lines (Fig. 2a, b). However, there were no significant changes in the expression of Fas, TRAIL, and NKG2D on AT3ova tumor cells following MEKi treatment in vivo (Supplementary Fig. 1A). Given that pronounced effects of MEKi were seen on MHC-I expression (Fig. 2a, b), we next explored the effects of MEKi-induced MHC-I expression on tumor cells and subsequent T-cell responses. To evaluate this, we investigated the in vitro effector function of ovalbumin-specific CD8+ OT-I T cells co-cultured with trametinib (MEKi) treated AT3ova tumor cells (Fig. 3a, b). When AT3ova cells were treated with MEKi, MHC-I expression was significantly upregulated (one-way ANOVA, $P < 0.01$) by cell intrinsic

mechanisms on the tumor cells, potentially enhancing their ability to present antigen to CD8+ OT-I ovalbumin-specific T cells (Fig. 3a). We next investigated the effector function of these CD8+ OT-I T cells and found that when co-cultures were performed with MEKi pre-treatment of AT3ova (priming), there was significantly enhanced (Student's *t*-test; $P < 0.0001$) IFNγ cytokine production by these CD8+ OT-I T cells (Fig. 3b). In contrast, in the continuing presence of MEKi in the co-culture, T-cell function was significantly inhibited (one-way ANOVA; $P < 0.0001$), as evidenced by the lack of IFNγ production (Fig. 3b). Taken together, this data suggest that MEKi increases tumor immunogenicity, however at the same time impairs T-cell effector function. Interestingly, fluorescence activated cell sorting (FACS) analysis of 4-1BB and OX-40 expression on CD8+ OT-I T cells (Supplementary Fig. 2A) and CD4+ OT-II T cells (Supplementary Fig. 2B) following co-culture with trametinib pre-treated (primed) AT3ova cells, revealed an increased expression of these co-stimulatory markers on T cells, leading us to investigate whether immune agonist antibodies targeting 4-1BB and OX-40 could aid in recovering this loss of functional T-cell activity.

**Kinetics of MEKi on T cells in vivo.** In order to explore the effect of MEKi on TIL proportions, cytokine production, and proliferation, we performed a time course FACS analysis of AT3ova tumors from mice treated with MEKi daily over the course of 9 days. MEKi appeared to have an inhibitory effect on T-cell effector function early in the treatment response from days 2–6, where a significant reduction (Student's *t*-test; $P < 0.05$) in the frequency and function of T cells was observed (Fig. 3c; Supplementary Fig. 3A, E). Decreased frequency of CD8+ T cells and both populations of CD4+ T cells; helper (FOXP3−) and T regulatory cells (Tregs; FOXP3+) were observed (Fig. 3c). At day 4 post treatment, significant decreases (Student's *t*-test; $P < 0.05$) in CD8+ and CD4+ T-cell cytokine production (IFNγ) (Fig. 3d; Supplementary Fig. 3B) and proliferation (Ki67) (Fig. 3e; Supplementary Fig. 3C), was evident in the MEKi-treated group compared with the vehicle control-treated group. In contrast to early time points, we observed that T-cell cytokine production and proliferation, but not TIL frequency (Fig. 3c), rebounded in MEKi-treated tumors by day 9 (Fig. 3d, e). However, we believe that this may in part be explained by the larger size of vehicle-treated tumors at later time points, leading to a more immuno-suppressive tumor microenvironment, which inhibits T-cell activity to a greater extent than MEKi. Analysis of innate immune subsets such as NK cells and NK T cells revealed no changes in frequency (CD3+ NK1.1+, CD3− NK1.1+), maturation (CD11b+ CD27−), or effector function (Granzyme B+) following MEKi (Supplementary Fig. 4A, B). As such, the focus of subsequent experiments was on the effects of MEKi on T cells specifically. To confirm that the observed effects of MEKi on T-cell frequencies were not due to modulation of the frequency of other immuno-suppressive immune subsets, we next quantified absolute numbers of various immune cell subsets infiltrating tumors. These experiments revealed that the number of CD45+ cells remained constant in the vehicle- and MEKi-treated groups at day 4 (Supplementary Fig. 5A). Furthermore, this analysis showed that only CD8+ and CD4+ T-cell numbers were reduced following MEKi (Supplementary Fig. 5B), while the numbers of other immunosuppressive subsets including macrophages (CD11b+ F4/80+ TAMs) and MDSCs (CD11b+ Ly6C+/Ly6G+) remained constant (Supplementary Fig. 5B). Both CD4+ FOXP3− T cells and CD4+ FOXP3+ Tregs showed an overall decrease in cell numbers (Supplementary Fig. 5C). Analysis of tumor-specific T cells using the H2Kb ovalbumin (SIINFEKL) tetramer revealed that MEKi similarly reduced the number of both tumor antigen-

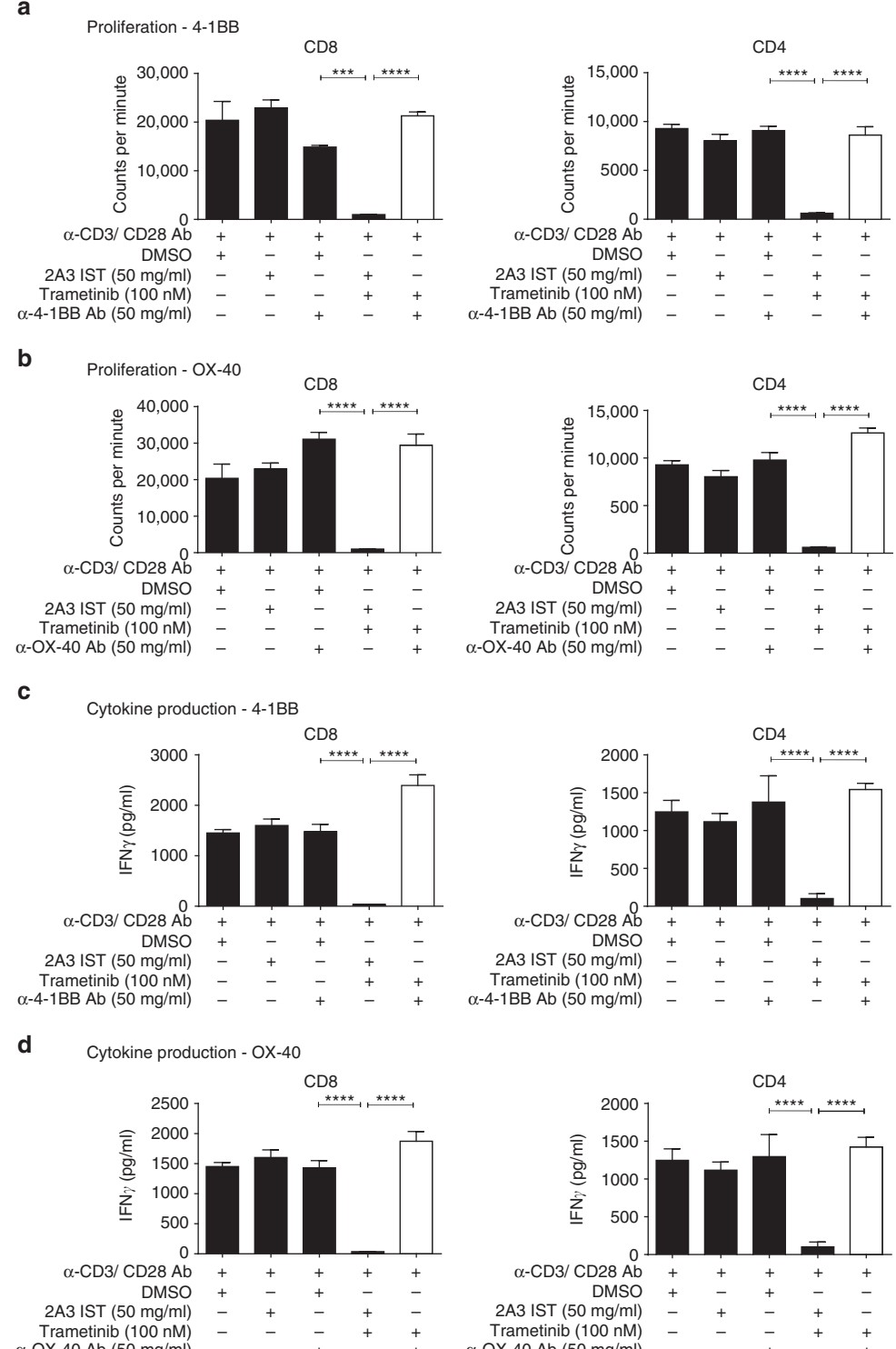

**Fig. 4** Agonist immunotherapy treatment rescues inhibition of mouse T-cell function induced by MEK inhibition. Purified CD8[+] or CD4[+] murine T cells were stimulated with α-CD3 (1 μg/ml) and α-CD28 (0.5 μg/ml) antibodies and treated with vehicle (DMSO), 2A3 isotype, α-4-1BB or α-OX-40 antibody, trametinib alone, or combination of trametinib and agonist antibody. **a**, **b** Cell proliferation was measured by 3H-thymidine incorporation (added at 48 h) after 72 h of incubation with treatments. Proliferation in counts per minute (CPM) was measured for CD4[+] and CD8[+] T cells for **a** α-4-1BB antibody and **b** α-OX-40 antibody combinations. **c**, **d** IFNγ cytokine production (pg/ml) was measured from 72 h cultured supernatants via CBA analysis of **c** α-4-1BB antibody and **d** α-OX-40 antibody combinations in both CD8[+] and CD4[+] T cells, respectively. Experiments were performed in quadruplicate and is representative of 2–3 independent repeats. Controls groups are duplicated between panels as these experiments were performed concurrently. Data are presented as mean ± SEM. *P*-values represent one-way ANOVA and post hoc Fisher's LSD tests. *$P < 0.05$, **$P < 0.01$, ***$P < 0.001$, ****$P < 0.0001$

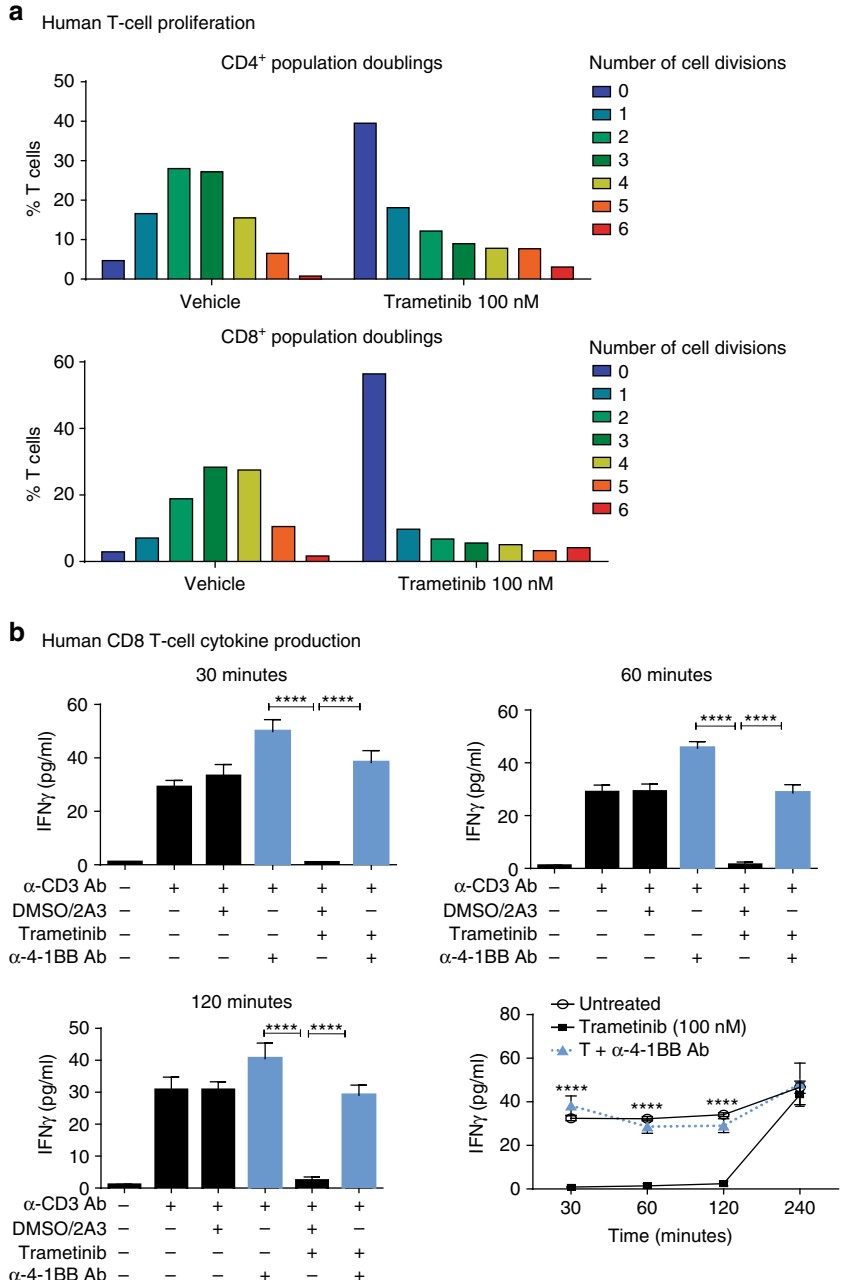

**Fig. 5** Agonist immunotherapy rescues human T-cell effector function following MEK inhibition treatment. Purified naive CD4+ and CD8+ T cells were isolated from donor human PBMCs and used for proliferation and cytokine production assays. **a** Carboxyfluorescein succinimidyl ester (CFSE) dilution FACS analysis of human CD4+ and CD8+ T-cell population doublings after 96 h of stimulation with α-CD3 antibody in the presence of vehicle or trametinib (100 nM). Numbers represent the number of cell divisions. **b** IFNγ production of human CD8+CD45RA+ T cells following 16 h of α-CD3 (1 μg/ml) antibody pre-stimulation and treatment: unstimulated, untreated (α-CD3 only), trametinib (100 nM) only, α-4-1BB (50 μg/ml) antibody alone, or combination of trametinib and agonist antibody. Experiment was performed in quadruplicate and is representative of 2–3 independent repeats. Data are presented as mean ± SEM. *P*-values represent unpaired *t*-tests and one-way ANOVA, post hoc Fisher's LSD tests. *$P < 0.05$, **$P < 0.01$, ***$P < 0.001$, ****$P < 0.0001$

specific CD8+ T cells (tetramer positive) and CD8+ T cells recognizing unknown antigens (tetramer negative) (Supplementary Fig. 5C). This indicates that the MEKi-induced inhibition is a global effect across all CD8+ T cells. Taken together, our data demonstrate clear early dampening of the initial T-cell immune response at days 0–6 post therapy, which may potentially reduce TIL anti-tumor functionality and the overall efficacy of treatment.

**Rescue of MEKi-mediated T-cell dysfunction in vitro.** We undertook cell proliferation and functional assays utilizing

purified mouse CD4+ and CD8+ T cells to test the hypothesis that impaired T-cell activity due to MEKi could be rescued via the addition of agonist antibodies. Purified T cells were stimulated with α-CD3/CD28 and treated for 72 h with either dimethyl sulfoxide (DMSO), 2A3 isotype, trametinib, α-4-1BB, α-OX-40, or the trametinib/agonist antibody combinations in vitro. In this experiment, we found that MEKi significantly reduced (one-way ANOVA; $P < 0.001$) both the proliferation (Fig. 4a, b) and IFNγ production (Fig. 4c, d) of both CD8+ and CD4+ T cells. Notably, the addition of either α-4-1BB (Fig. 4a) or α-OX-40 (Fig. 4b) significantly enhanced (one-way ANOVA; $P < 0.0001$)

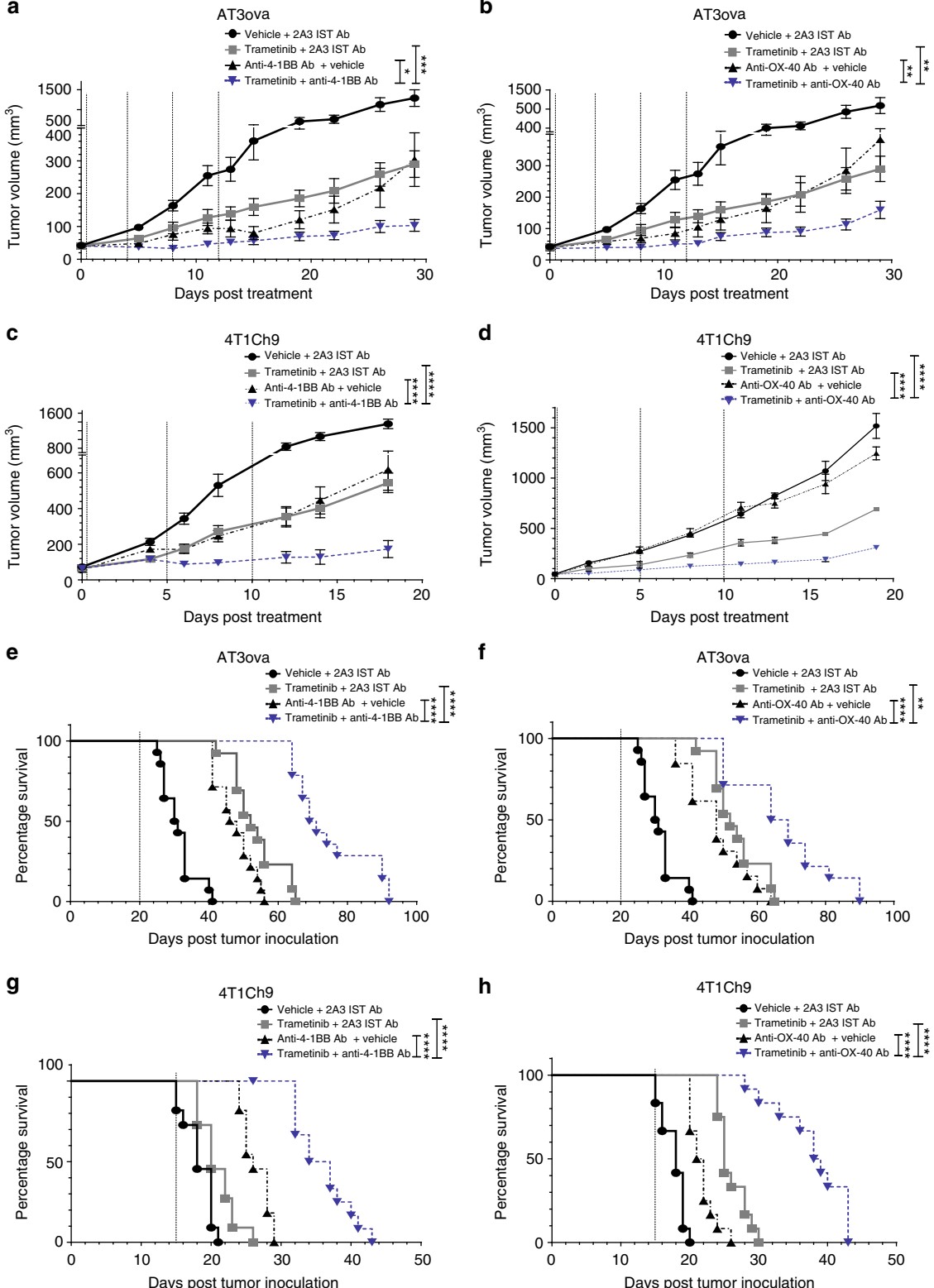

**Fig. 6** Combined MEK inhibition and agonist immunotherapy leads to enhanced efficacy of treatment against established TNBCs. Mice ($n = 6$ per group) bearing established AT3ova **a**, **b**, **e**, **f** or 4T1Ch9 **c**, **d**, **g**, **h** tumors were treated with vehicle or trametinib via daily oral gavage for 20 days (AT3ova) or 15 days (4T1Ch9) and either 2A3 isotype control, α-4-1BB, or α-OX-40 antibody alone; three doses (4T1Ch9) or four doses (AT3ova) via IP injection on days 0, 4, 8, and 12 or combination of trametinib and agonists. **a**–**d** Tumor growth volume (mm³) and **e**–**h** survival ($n = 12$) were monitored. End point was determined as when tumors reached an ethical limit of 1400 mm³. Experiments are a representative of $n = 2$–3 replicates, with pooled mouse numbers for survival (mean tumor volume ± SEM). Controls groups are duplicated between panels as these experiments were performed concurrently. *P*-values represent two-way ANOVA, post hoc Tuckey's tests for tumor growth, and log ranked (Mantel–Cox) test for survival proportions. *$P < 0.05$, **$P < 0.01$, ***$P < 0.001$, ****$P < 0.0001$

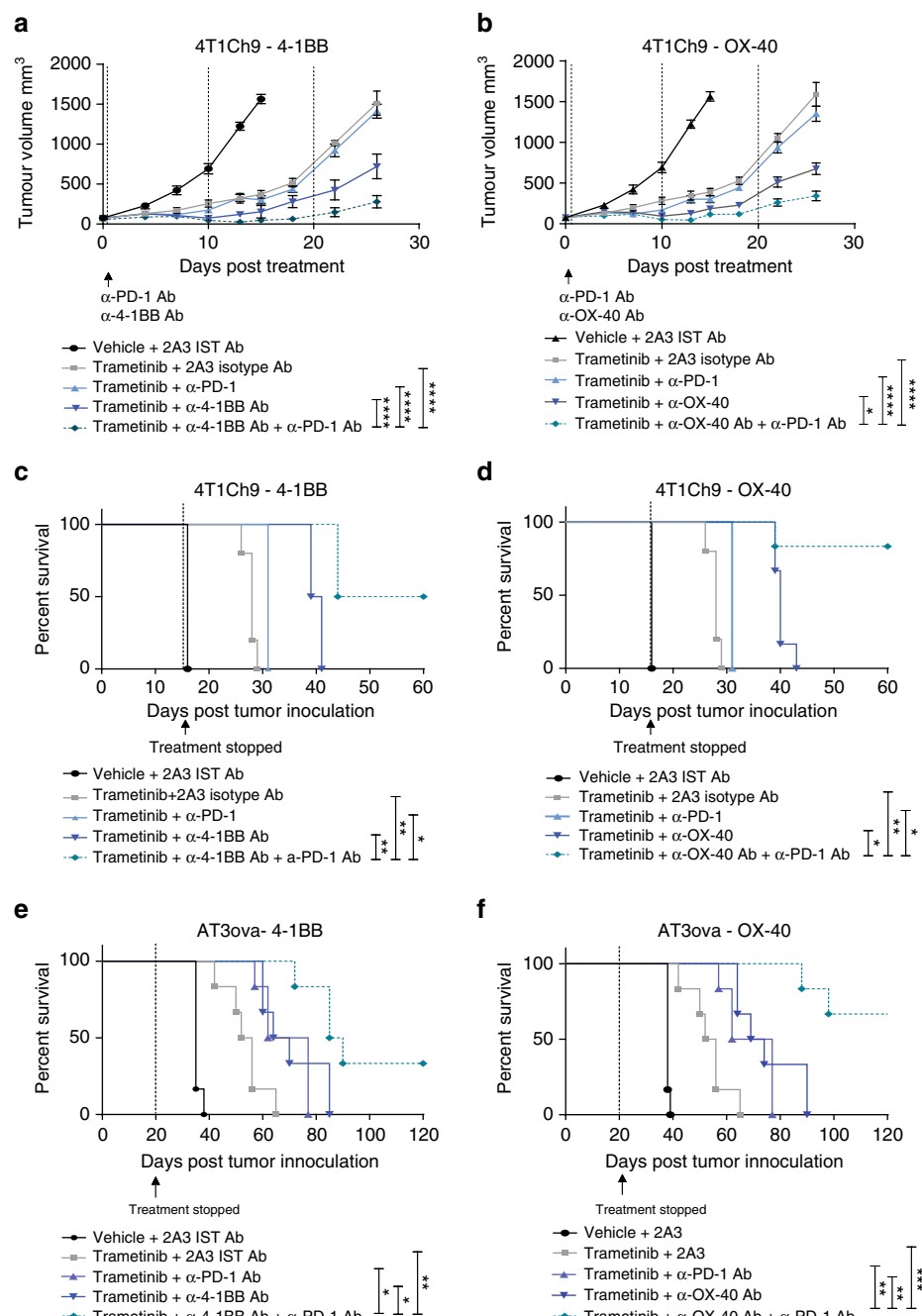

**Fig. 7** Triple combination of MEK inhibition and agonist immunotherapy plus anti PD-1 checkpoint blockade leads to enhanced efficacy in AT3ova and 4T1Ch9 tumor bearing mice. Mice ($n = 6$ per group) bearing established 4T1Ch9 (**a–d**) or AT3ova (**e, f**) tumors were treated with a double combination of trametinib via daily oral gavage for 20 days (AT3ova) or 15 days (4T1Ch9) and either α-4-1BB, α-OX-40, or α-PD-1 antibody or triple combination of trametinib and either agonist (α-4-1BB, α-OX-40) in combination with α-PD-1 antibody; three doses (4T1Ch9) or four doses (AT3ova) via IP injection on days 0, 4, 8, and 12 or combination of trametinib and agonists. **a, b** Tumor growth volume ($mm^3$) in the 4T1Ch9 model and survival ($n = 6$) in the 4T1Ch9 (**c, d**) and AT3ova (**e, f**) were monitored. End point was determined as when tumors reached an ethical limit of 1400 $mm^3$ (mean tumor volume ± SEM). Controls groups are duplicated between panels as these experiments were performed concurrently. $P$-values represent two-way ANOVA, post hoc Tuckey's tests for tumor growth, and log ranked (Mantel–Cox) test for survival proportions. *$P < 0.05$, **$P < 0.01$, ***$P < 0.001$, ****$P < 0.0001$

proliferation of both CD4+ and CD8+ T cells in the presence of MEKi, overcoming the detrimental effect of MEKi. Similarly, these agonistic antibodies significantly (one-way ANOVA; $P < 0.0001$) restored the production of IFNγ (Fig. 4c, d) from these T cells, in combination with MEKi. We next tested whether agonist immunotherapy could overcome diminished T-cell activity in the context of MEKi in human T cells, where we found that MEKi significantly diminished (one-way ANOVA; $P < 0.05$)

the proliferation of both CD4+ and CD8+ T cells, up to 96 h post treatment (Fig. 5a). Cytokine analysis also revealed that MEKi significantly reduced (one-way ANOVA; $P < 0.0001$) the production of IFNγ by human CD8+ T cells (Fig. 5b). Similarly to as shown in mouse CD8+ T cells, this inhibitory effect on effector function was significantly prevented (one-way ANOVA; $P < 0.0001$) by the addition of the α-4-1BB agonist antibody in human CD8+ T cells (Fig. 5b). Taken together, this data highlight

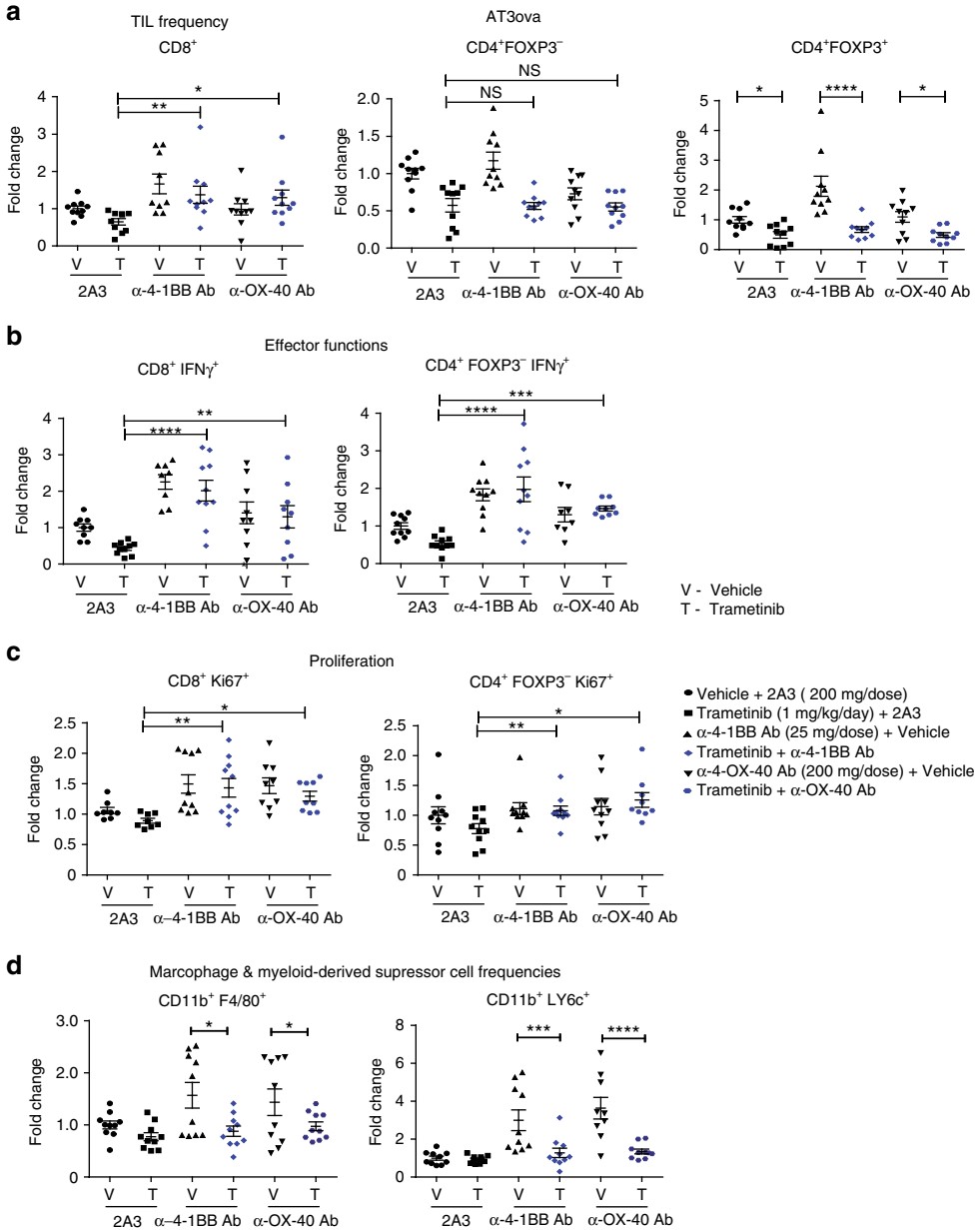

**Fig. 8** Agonist immunotherapy rescues T-cell effector functions in the presence of MEK inhibition. Mice bearing established AT3ova tumors were treated with vehicle, trametinib plus isotype control antibody and either α-4-1BB or α-OX-40 antibody alone; IP injection on day 0, or combination of trametinib and agonists. Changes in TIL populations (CD8[+], CD4[+] FOXP3[−], CD4[+] FOXP3[+] T cells, macrophage, and MDSCs) were determined ex vivo by FACS analysis 4 days post treatment. **a** TIL frequency as a proportion of CD45[+] live cells, **b** IFNγ cytokine production by T cells, **c** proliferation of T cells measured by Ki67 expression, and **d** frequency of macrophage (CD11b[+], F4/80[+]) and MDSC (CD11b[+], Ly6c[+]) subsets. Values were normalized to vehicle controls in each experiment. Data are expressed as fold change ± SEM for the number of positive cells and represents $n = 5–10$ mice per group, pooled from two independent experiments. *P*-values represent one-way ANOVA, post hoc Fisher's LSD tests.*$P < 0.05$, **$P < 0.01$, ***$P < 0.001$, ****$P < 0.0001$

the ability of these agonist antibodies to prevent the impairment of T-cell proliferation and cytotoxic effector function mediated by MEKi. The results from these studies led us to evaluate these MEKi agonist immunotherapy combinations in vivo.

**Combination therapy rescues T-cell effector function in vivo.** We utilized two murine TNBC models, AT3ova (Fig. 6a, b, e, f) and 4T1Ch9 (Fig. 6c, d, g, h) to investigate the potential efficacy of combination therapy with MEKi and either α-4-1BB or α-OX-40 antibodies. Analysis at the genomic level (Supplementary Fig. 6A) and the transcriptional level (Supplementary Fig. 6B) confirmed Ras/MAPK pathway activation in both the

AT3ova and the 4T1Ch9 cell lines. Once tumors were established (~35–60 mm³), mice were treated with a vehicle control or trametinib and isotype control, or either α-4-1BB or α-OX-40 antibody alone, or a combination of trametinib and either isotype control, α-4-1BB, or α-OX-40 antibody. Consistent with our previous observations, some anti-tumor activity was observed with trametinib administered alone[15]. However, combined treatment of MEKi with α-4-1BB antibody resulted in significantly enhanced (two-way ANOVA; $P < 0.001$) inhibition of tumor growth in both the AT3ova and 4T1Ch9 models (Fig. 6a, c), and significantly prolonged (log ranked (Mantel–Cox); $P < 0.0001$) the survival of mice, compared to either single agent alone (Fig. 6e, g). Similarly, combined

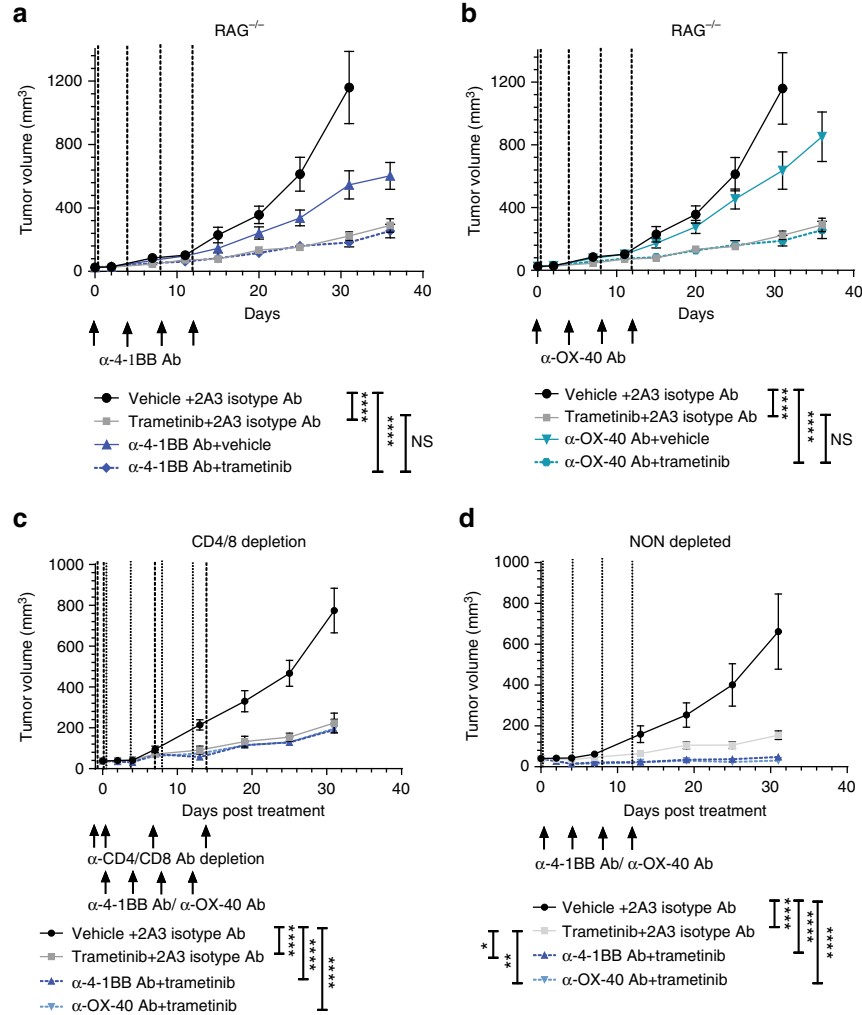

**Fig. 9** MEK inhibition and agonist combination therapy in RAG and T-cell depletion models demonstrates the functional role of T cells in enhancing therapeutic responses. **a**, **b** RAG$^{-/-}$ mice ($n = 8$ per group) bearing established AT3ova tumors were treated with vehicle or trametinib via daily oral gavage for 20 days (AT3ova) and either 2A3 isotype control, α-4-1BB, or α-OX-40 antibody alone; four doses (AT3ova) via IP injection on days 0, 4, 8, and 12 or combination of trametinib and agonists. **c** WT C57BL/6 mice ($n = 6$ per group) bearing established AT3ova tumors were concurrently depleted of CD4 and CD8 T cells using depletion antibodies (200 μg/dose) on day −1, 0, 7, and 14 of treatment. **d** Matched non-depleted control groups of WT C57BL/6 mice ($n = 6$ per group) bearing established AT3ova tumors were also administered the same treatment schedule as described above for single and combination therapies. **a–d** Tumor growth volume (mm$^3$) was monitored. Controls groups are duplicated between panels as these experiments were performed concurrently. Vehicle and trametinib arms are duplicated in **a** and **b**. *P*-values represent two-way ANOVA, post hoc Tuckey's tests for tumor growth. \*$P < 0.05$, \*\*$P < 0.01$, \*\*\*$P < 0.001$, \*\*\*\*$P < 0.0001$

trametinib and α-OX-40 antibody treatment, significantly (two-way ANOVA; $P < 0.01$) enhanced therapeutic efficacy in terms of delayed tumor growth (Fig. 6b, d) and extended survival (log ranked (Mantel–Cox); $P < 0.01$) in both models, compared to single agent therapy (Fig. 6f, h). We have previously shown that α-PD-1 can enhance the therapeutic effects of MEKi[15] and thus we compared the effects observed with α-4-1BB and α-OX-40. Strikingly, in the 4T1Ch9 model, the efficacy of either α-4-1BB/MEKi or α-OX-40/MEKi was significantly greater (two-way ANOVA; $P < 0.0001$) than the combination of α-PD-1 and MEKi (Fig. 7a, b). Furthermore, since we have previously observed increased expression of PD-1 on tumor-infiltrating CD8$^+$ T cells following α-4-1BB/α-OX-40 treatment, we next investigated the efficacy of combining MEKi with both an agonist immunotherapy regimen (α-4-1BB/α-OX-40) and blockade with α-PD-1 antibody. We found that the survival of mice was significantly (two-way ANOVA; $P < 0.05$) improved (Fig. 7c–f), particularly in the

α-OX-40 combination, with roughly 70% of AT3ova bearing mice surviving over 120 days (Fig. 7f). Overall, this data demonstrate the therapeutic potential of agonist immunotherapy and checkpoint blockade to significantly enhance anti-tumor responses through the restoration of MEKi-mediated early loss of T-cell function.

**Immune mechanism of combination therapy in vivo**. Following our observation of reduced anti-tumor T-cell responses induced by trametinib in vivo (Fig. 3c–e), we next assessed the immunological mechanisms underlying the enhanced anti-tumor immune responses mediated by the combination of MEKi and agonistic antibodies. We undertook extensive FACS analysis on ex vivo AT3ova tumors at day 4 of treatment based upon our time course analysis shown in Fig. 3c–e. MEKi treatment alone decreased the proportion of CD8$^+$, CD4$^+$ FOXP3$^-$, and

CD4+ FOXP3+ cells (Fig. 8a), and reduced IFNγ production of CD8+ T cells (Fig. 8b) as observed previously at this time point. However, the addition of either α-4-1BB or α-OX-40 antibody, resulted in restored CD8 T-cell frequency (Fig. 8a; Supplementary Fig. 7A), significantly (one-way ANOVA; $P < 0.01$) increased IFNγ production by CD8+ T cells (Fig. 8b; Supplementary Fig. 7B), restoring the loss of IFNγ production mediated by MEKi. Furthermore, we observed that proliferation was also enhanced in CD8+ T cells following α-4-1BB or α-OX-40 combination therapy (Fig. 8c; Supplementary Fig. 7C). Similarly, α-4-1BB and α-OX-40 antibodies significantly (one-way ANOVA; $P < 0.01$) increased IFNγ production from CD4+ FOXP3− T cells (Fig. 8b) and enhanced proliferation of these T cells (Fig. 8c). Interestingly, anti-4-1BB/anti-OX-40 enhanced the frequency of macrophages (CD11b+ F4/80+) and MDSCs (CD11b+ Ly6C+), which was reduced following treatment with MEKi (Fig. 8d). These effects of increased TIL proportions, CD4+, and CD8+ T-cell proliferation and cytokine production were sustained long term, as evidenced by significantly enhanced TIL activity observed at later time points (Supplementary Figs. 8A–C, 9A–C). Moreover, we observed increased expression of T-cell markers such as Tbet+ and CCR7+, associated with a T$_H$1-like

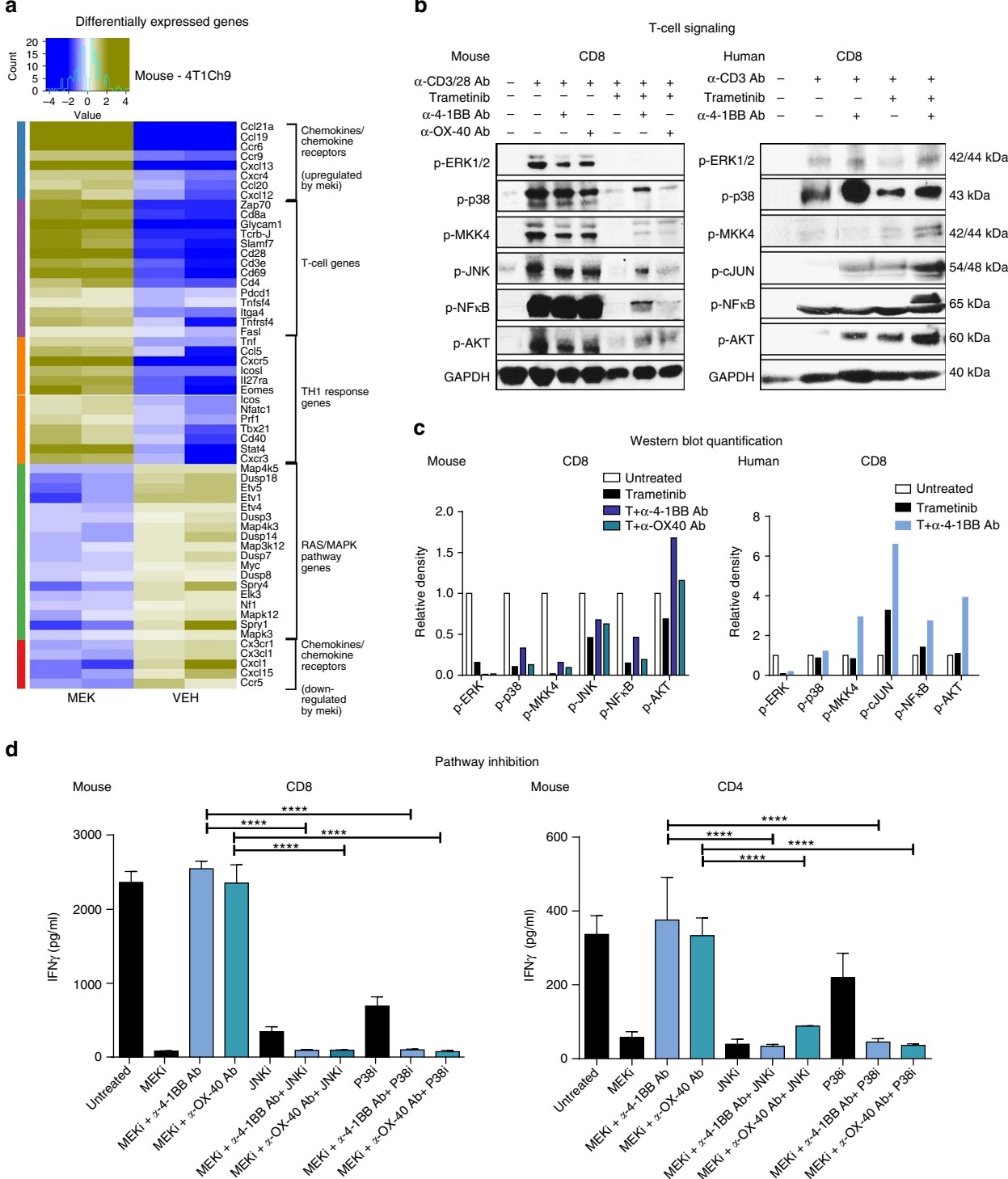

memory response for both α-4-1BB and α-OX-40 antibody combinations with MEKi, at later time points (Supplementary Figs. 8C, D, 9C, D). To demonstrate the importance of T cells in the therapeutic effects observed with MEKi and α-OX-40/ α-4-1BB, we undertook further studies in RAG$^{−/−}$ mice (which lack T and B cells) and in wild-type mice depleted of CD4$^+$/CD8$^+$ T cells. We show that in both RAG$^{−/−}$ (Fig. 9a, b) and CD4$^+$/CD8$^+$ T-cell depleted wild-type mice (Fig. 9c, d), the addition of agonist immunotherapies (α-4-1BB and α-OX-40) had no additional effects to MEKi alone (Fig. 9a–d).

**MEKi alters immune pathway gene expression**. In order to further understand the impact of MEKi on the immune response in vivo, we investigated, in an unbiased manner, the differential expression of genes in vehicle-treated compared with trametinib-treated 4T1Ch9 tumors (Fig. 10a). We analyzed gene expression at day 7 post treatment. This time point was determined from time course FACS data at day 7, where we observed that TILs were beginning to recover from MEKi treatment (Fig. 3c–e). In total, there were 3932 genes significantly altered between MEKi and vehicle ($P < 0.05$, fold change > 1.5, and FDR < 10%). As expected, we saw the significant downregulation ($P < 0.05$, fold change >1.5, and FDR < 10%) of Ras/MAPK pathway-associated genes compared with vehicle control treatment (Fig. 10a). We found that immune genes such as *CD3ε, CD4, CD8α, CD28, TCR-β*, TNFs, and various chemokine receptors (CXCR4, CCR6,9, CXCL12,13, CCL19-21) were upregulated in the MEKi-treated tumors (Fig. 10a; Supplementary Data 1) consistent with the rebound in TIL infiltrate observed at day 7. Additionally, we observed many genes associated with a Th1 signature (*TNF, CCL5, CXCR5, CXCR3, Tbx21, Eomes, Icos, Stat4, Nfatc, Il27ra, Tnfsf4, CD40, Prf1*) that were significantly upregulated ($P < 0.05$, fold change > 1.5, and FDR < 10%) following MEKi. Notably, GeneGo pathway analysis[33] of these differentially expressed genes demonstrated that numerous immune response pathways such as PD-1, CTLA-4 checkpoints, T-cell receptor signaling, and several chemokine/chemotaxis factors were significantly ($P < 0.05$, fold change > 1.5, and FDR < 10%) enriched as a result of MEKi (Supplementary Fig. 10A; Supplementary Data 2). Gene set enrichment analysis[34] also confirmed that immune response and T-cell activation signatures were significantly ($P < 0.05$, fold change > 1.5, and FDR < 10%) enriched in tumors treated with MEKi, supporting the hypothesis of increased immunogenicity (Supplementary Fig. 10B, C; Supplementary Data 3, 4). Interestingly, we observed that NFκB, p38/JNK pathway genes were significantly ($P < 0.05$, fold change > 1.5, and FDR < 10%) enriched following MEKi treatment

(Supplementary Fig. 10A, D, E; Supplementary Data 5, 6). These pathways are known to be regulated downstream of MAPK and could act as alternative signaling pathways to the classical canonical Ras/MAPK/MEK1/2 activation in T cells. This led us to further evaluate these pathways in primary mouse and human T cells.

**T-cell signaling pathways induced by combination therapy**. We next investigated if the activation of NFκB and/or p38/JNK pathways was the mechanism by which T cells were activated as a result of α-4-1BB and α-OX-40 stimulation in the context of MEKi, in both mouse and human T cells (Fig. 10b). We performed western blot analysis for key targets of the 4-1BB and OX-40 signaling pathways that included; p38, NFκB, and JNK pathway genes, shortlisted based on analyses of the gene expression data from 4T1Ch9 mouse tumors (Supplementary Fig. 10A, D, E). As expected, we found that MEKi alone resulted in a reduction in ERK1/2 phosphorylation (the immediate downstream target of MEK1/2), in both mouse and human CD8$^+$ T cells (Fig. 10b). Quantitative analysis (densitometry) was undertaken for mouse and human western blots (Fig. 10c). We observed that α-4-1BB or α-OX-40 immunotherapy in combination with MEKi enhanced the phosphorylation of p38/JNK/NFκB pathway targets compared to MEKi alone but did not restore ERK1/2 phosphorylation (Fig. 10b, c; Supplementary Fig. 12). In order to validate that both p38 and JNK pathways were crucial for the rescue effect observed following agonist therapy, we undertook a cytometric bead array (CBA) analysis using inhibitors for both p38 and JNK. The results demonstrated that inhibiting both JNK and p38 following MEKi and agonist combination therapy, significantly (one-way ANOVA; $P < 0.0001$) diminished cytokine production by CD4$^+$ and CD8$^+$ T cells (Fig. 10d). These results suggest that restored T-cell signaling mediated by α-4-1BB or α-OX-40 occurs through ERK1/2 independent routes. We also observed significant upregulation of p-AKT, which suggests that inhibition of the Ras/MAPK pathway leads to re-direction of signaling through the PI3K pathway. The proposed mechanism of action for signaling in combination therapy is illustrated in Supplementary Fig. 11.

**Discussion**

It is evident that more effective strategies for the treatment of TNBCs are urgently needed, given the lack of therapeutic targets and durable treatment options currently available. TNBC is clinically recognized as having the poorest prognosis of all breast cancer subtypes[12]. The characteristic genomic instability and high mutational loads observed in TNBC is thought to play a central role in promoting TIL recruitment through T-cell recognition of

**Fig. 10** Rescue of T-cell function by agonist antibodies occurs via activation of alternative MAPK signaling pathways independent of ERK. Ex vivo RNA extraction of trametinib-(1 mg/kg/day) treated 4T1Ch9 or vehicle-treated tumors was undertaken to perform Affymetrix microarray analysis. **a** Heat map representing upregulated and downregulated differentially expressed genes between vehicle- and MEKi-treated 4T1Ch9 tumors (duplicates/condition) based on adjusted *P*-value of <0.05. Hierarchical clustering was performed. **b** Mouse and human T-cell signaling was analyzed using purified CD8$^+$ mouse T cells isolated from mouse spleen and activated for 16 h with α-CD3 (1 μg/ml) antibody. Treatment groups include: unstimulated or stimulation with α-CD3/CD28 antibody (0.5 μg/ml in mouse), trametinib (10 nM), α-4-1BB (50 μg/ml) or α-OX-40 (50 μg/ml) antibody alone, or combination of trametinib and agonist antibody for an incubation period of 72 h. Human T-cell signaling was analyzed using purified CD45RA$^+$ CD8$^+$ human T cells isolated from human PBMCs and activated for 16 h with α-CD3 (OKT3; 1 μg/ml) antibody. Treatment groups were as above, dosed for an incubation period of 5 min. **b** Western blot analysis was performed for phosphorylated proteins of T-cell signaling pathways; p-ERK (42/44 k Da), p-p38 (43 kDa), p-MKK3/6 (38/40 kDa), p-MKK4 (44 kDa), p-AKT (60 kDa), p-JNK (54 kDa), c-JUN (48 kDa), p-NFκB (65 kDa), and GAPDH (40 kDa). **c** Western blot quantitation based on relative density, normalized to untreated control. **d** CBA analysis of IFNγ production by α-CD3/CD28 antibody- (1 or 0.5 μg/ml) activated mouse CD4 and CD8 T cells following MEKi (100 nM) and agonist rescue (50 μg/ml) with inhibition of JNK and P38 pathways using P38 inhibitor (10 μM) and JNK inhibitor (10 μM) following 72 h of treatment. Experiment was performed in quadruplicate and is representative of 2–3 independent repeats. Data are presented as mean ± SEM. *P*-values represent one-way ANOVA, post hoc Fisher's LSD tests. \**P* < 0.05, \*\**P* < 0.01, \*\*\**P* < 0.001, \*\*\*\**P* < 0.0001

mutant neo-antigen peptides[31, 35]. Higher TIL levels are strongly associated with better outcomes in TNBC[7, 11, 12]. However, thus far α-PD-1 or α-PDL-1 antibodies have shown only modest activity as monotherapies[36, 37]. We have previously shown that genomic alterations in the Ras/MAPK pathway correlate with lower TIL infiltrate in TNBC[15], raising the hypothesis that targeting MEK may enhance TIL infiltrate and thereby potentiate the efficacy of immunotherapy with the ultimate aim of improving survival.

Although the evidence suggests that targeting MEK may increase tumor immunogenicity, potentially allowing for greater TIL infiltrate, MEKi has also been shown to adversely affect anti-tumor immunity[23, 25, 38]. Herein, we show in models of TNBC that MEKi has early detrimental effects on T-cell frequency, and function in terms of proliferation and cytokine production. Since MEK signaling occurs downstream of TCR activation (Supplementary Fig. 11), it is likely that MEKi affects these parameters concurrently. These effects are clearly functionally important, as we demonstrate that agonist immunotherapies can prevent this inhibitory effect on T-cell signaling and thereby improve anti-tumor efficacy. To support this, we show that the anti-tumor efficacy of MEKi is similar in T-cell depleted (either RAG or antibody depleted) or non-depleted mice. This is consistent with our data indicating that MEKi inhibits the effectiveness of anti-tumor T-cell responses by suppressing T-cell effector functions. Our experiments also revealed that, as expected, the addition of immune agonists (α-4-1BB or α-OX-40) enhanced anti-tumor effects in combination with MEKi in T-cell replete, but not in T-cell depleted mice.

Our observations of increased immunogenicity following MEKi are in accordance with the findings of Liu et al.[21], who showed that HLA I/II (MHC-I/II) as well as melanoma-associated tumor antigens were upregulated following MEKi. The ability of MEKi to increase tumor immunogenicity supports clinical data currently emerging in solid tumors previously unresponsive to checkpoint blockade monotherapy[36, 37]. Interestingly, the effects of MEKi on MHC-I expression were most pronounced in the context of IFNγ. While the molecular targets of MEKi leading to increased MHC-I expression remain undefined, this suggests that it may interact with signaling downstream of the IFNγ receptor. Nevertheless, this data clearly support the concept that combining MEKi with immunotherapy enhances IFNγ production, which has the potential to significantly enhance MHC-I expression and, consequently, tumor cell immunogenicity. Although we have focussed upon increased expression of MHC-I as a mechanism of increased tumor immunogenicity in this study, our findings of increased expression of other tumor ligands such as Fas, TRAIL, and NKG2DL (RAE-1) in vitro suggests that further investigation into the relevance of these pathways in the observed therapeutic effects of MEKi is warranted. Taken together, our findings provide strong evidence that MEK pathway activation in the tumor drives inhibition of immunogenicity, given that inhibiting MEK increases MHC-I, and thus improves TIL recognition of tumor antigens, leading to enhanced cytotoxic responses. Moreover, we show that this initial loss of T-cell function is of significant detriment to the overall anti-tumor efficacy, as combining agonistic antibodies with MEKi resulted in significantly enhanced anti-tumor immune responses and considerably prolonged survival in vivo, in two distinct immunocompetent mouse models of TNBC; modeling both high and low TIL settings. Furthermore, combination with α-PD-1 checkpoint blockade and agonist immunotherapy with MEKi in the AT3ova and 4T1Ch9 models, demonstrated significantly improved survival outcomes, compared to MEKi and agonist immunotherapy alone. Intriguingly, the triple combination had significant survival benefits in the low TILs model (4T1Ch9),

which was resistant to the MEKi/α-PD-1 double combination therapy. These results are similar to the findings of Moreno et al.[39], where the authors showed that a quadruplet combination of a BRAFi, MEKi, either α-PD-1 or α-PDL-1 and either α-4-1BB or α-OX-40 significantly delayed tumor growth in BRAF mutant melanoma models. However, the effects of these combinations on survival were not shown, and the toxicity associated with combining multiple immunotherapies and targeted inhibitors together was not discussed[39]. Our study provides insight into the mechanism underlying the efficacy of these combination therapies.

This observation of T-cell inhibition with MEKi is consistent with the findings of Ebert et al.[23] who suggested that MEKi prevented T-cell priming in lymph nodes. Similarly, Hu-Lieskovan et al.[22] showed that cytokine production was decreased in vitro, although they observed that these effects were not as profound in vivo. However our study convincingly demonstrates that these inconsistencies are likely explained by the time points at which ex vivo tumors and TILs were analyzed in other studies[21, 22]. Both Boni et al.[25] and Vella et al.[24] found that MEKi alone inhibited T-cell proliferation, antigen-specific expansion, and cytokine production, while Liu et al.[21] reported that the effect on T cells was only transient in vitro. Interestingly, our studies in mouse and human T cells highlight some differences in the duration of signaling events, as well as some potential compensation in the pathways. This redundancy between mouse and human systems has previously been discussed[40]. Nonetheless, we observe the same inhibitory effect on T-cell signaling following MEKi in both mouse and human T cells, thus validating these findings. Strikingly, our study demonstrated that agonist antibodies that are currently in clinical trials can prevent this early suppressive effect. In another approach, Allegrezza et al.[41] demonstrated that the addition of IL-15 could overcome the immunosuppressive of MEKi on T-cell activity. However, the addition of agonistic antibodies presents a far more attractive approach to overcome the effects of MEKi, given the known difficulties associated with the toxicity profile of cytokine therapies[42].

In our study, we also show that MEKi reduces the frequency of Tregs. This could be related to the observation that MEKi suppresses TGF-β and IL-10 production in tumor cells, thus preventing the induction of Treg formation or immobilization of other immunosuppressive subsets such as myeloid-derived suppressor cells[43, 44]. While several studies have shown increased infiltration of TAMs and MDSCs following treatment[21, 22, 39], we show that these populations were unchanged following MEKi treatment alone. This is potentially due to the different effects of MEKi and the BRAFi used in these studies. Interestingly, in our study MEKi reduced the frequency of TAMs, MDSCs, and Treg subsets when combined with either anti-OX-40 or anti-4-1BB antibody, potentially owing to the fact that these subsets are driven by MAPK signaling[45, 46]. Collectively, while it is suggested that T cells may recover over the treatment course, it is evident that this initial loss of crucial and functionally important T-cell activity through MEKi is of significant detriment to the overall treatment efficacy. While our study conclusively shows that MEKi is detrimental to T-cell effector functions, the effect of MEKi on innate immune cells remains relatively unknown. In future studies, it will be interesting to characterise the requirement of MEK signaling in other immune cells subsets.

Our unbiased analysis of differentially expressed genes following MEKi supports the effect of Ras/MAPK pathway inhibition on immune cell function, but also highlights that other pathways are activated in response to reduced MEK activity, which may be further enhanced by α-OX-40/4-1BB therapy. This analysis revealed several chemokine/chemotactic factors that were

upregulated in response to MEKi. We observed some common genes such as *ICOSL*, *CXCL12*, and *CXCL13* in our analysis that were also found by Liu et al.[21]. Many of these chemokines and cytokines have been implicated in the immune response in breast cancer. For example, some breast cancer cell lines have been shown to express CXCR4-CXCL12, CCR6. CCR9, and CCL20, which are involved in promoting metastasis by enhancing tumor cell proliferation and migration[47]. Additionally, CCL19 has been found to activate T cells and CCL21, CXCR5-CXCL13 has been shown to regulate naive T-cell homing to secondary lymphoid organs[48]. Interestingly, some of these factors have also been reported to be involved in functions such as priming of dendritic cells[49] and promoting B-cell-mediated humoral responses[50]. As such, it may be of benefit to delineate the function of some of these factors in the tumor immune microenvironment in future studies. Ott et al. and Vella et al.[24] showed that MEKi promoted maturation of dendritic cells, thus impairing antigen uptake and processing, and ultimately cross presentation to T cells. Furthermore, other studies have shown that COX and PGE2 expression prevents dendritic cell accumulation and activation in tumors and that MEKi led to reduced expression of these factors[51]. Overall, this finding of enhanced chemoattraction of TILs to the tumor site following MEKi could provide us with potential candidates to target in future studies to enhance TIL infiltration in the low TILs patient setting.

In order to understand the mechanism underlying the improved efficacy observed with the addition of immune agonists α-4-1BB and α-OX-40 to MEKi treatment, we investigated what other compensatory signaling pathways may have been activated based on our pathway analysis studies. As such, we propose that re-direction of signaling from classical MEK1/2 activation occurs through alternative activation of the p38, NFκB, JNK pathways in 4-1BB and OX-40 signaling. We found that these pathways were upregulated following combination therapy, compared to MEKi alone in our western blot analysis. Moreover, using inhibitors for both p38 and JNK, we definitively show that the effects of these agonists in combination with MEKi are abrogated, demonstrating the functional importance of the p38/JNK pathways in the rescue of T-cell effector function. This is consistent with previous studies showing activation of these pathways downstream of 4-1BB/OX-40 activation[52, 53]. Indeed, it has been suggested that incomplete T-cell activation, maintained by hindered Ras/MAPK signaling, can be reversed by adequate co-stimulation with 4-1BB/OX-40[54]. We propose that these pathways are potentially activated downstream of AKT through PI3K pathway cross-talk in the absence of classical Ras/MAPK pathway MEK1/2 activation[52, 53]. Most notable was the significant increase in Akt phosphorylation induced by anti-4-1BB/anti-OX-40 in the presence of MEKi. Interestingly IL-15 is known to activate the PI3K/Akt pathway, which may explain the enhanced therapeutic effects observed by Allegrezza et al.[41] Thus, we suggest that activation of these alternative pathways including PI3K/Akt with agonist rescue may be responsible for the restored proliferation and cytokine production of T cells observed in the combinations. This knowledge may provide further avenues of investigation in our attempts to enhance anti-tumor immune effects in TNBC patients.

In summary, we conclude that targeting the MEK pathway with immunotherapy combinations in TNBC has synergistic effects, particularly as a significant proportion of human TNBCs are MEK activated and MEK activation is associated with poorer clinical outcomes[15]. In the current study, we have demonstrated the significance of the adverse and detrimental effects of MEKi on early T-cell signaling in both the mouse and human setting. We definitively demonstrate that agonist immunotherapy can be effectively utilized to prevent this immunosuppressive effect, ultimately enhancing T-cell proliferation, effector T-cell cytotoxic activity, $T_H1$ responses, and tumor-TIL homing. Our data provide a strong rationale for the combined use of MEK-targeted therapies with agonist immunotherapy, which may be applicable in multiple cancer types; not exclusively MEK activated or immunogenic solid tumors, whereby MEKi can be utilized to prime immunogenicity of tumors. Furthermore, as both α-4-1BB and α-OX-40 antibodies are currently in early-phase clinical trials for many cancers, our results provide a solid rationale for combination with clinically available MEK inhibitors. Clinical trials with combination MEKi and checkpoint blockade are currently ongoing in metastatic TNBC (clinicaltrials.gov; NCT02322814) and this strategy could therefore increase the number of patients that could potentially benefit from immunotherapy.

## Methods

**Cell lines and culture**. Two murine TNBC cell lines, AT3ova of C57BL/6 origin and 4T1Ch9 of Balb/c background, were kindly donated to the laboratory by Dr. Trina Stewart (Griffith University) and Professor Robin Anderson (Peter MacCallum Cancer Centre), respectively. The AT3ova cell line (cultured with complete DMEM media) was generated by transducing the parental AT3 cell line with a retroviral vector, pMIG/MSCV-IRES-eGFP plasmid encoding membrane-bound chicken ovalbumin complimentary DNA (model antigen) peptide, tagged with GFP. The 4T1Ch9 cell line (cultured with complete RPMI media) is tagged with cherry. All cell lines used in these experiments have been verified to be mycoplasma negative by the Victorian Infectious Diseases References Lab (Melbourne, VIC, Australia).

**Drugs**. The MEK1/2 inhibitor Trametinib (MEKi; GSK12021101) was kindly provided by Glaxo-SmithKline/Novartis via a material transfer agreement (MTA). The drug was solubilized in DMSO at a 100 nM concentration for in vitro studies. For in vivo studies, trametinib was prepared in PEG 400/Solutol (1:4) solution at 1 mg/kg and administered daily via oral gavage with continuous dosing. Inhibitors for p38 (p38i; BIRB 796 (Doramapimod)) and JNK (JNKi; SP600125) were solubilized in DMSO at a concentration of 10 μM for the in vitro studies. Mouse recombinant IFNγ protein (Becton Dickinson, BD) was added at 5 ng/ml for in vitro assays. Mouse-specific antibodies for α-4-1BB (3H3 clone) and α-OX-40 (OX-86 clone) and isotype control (2A3) antibodies were purchased from BioXcell. For depletion studies, α-CD4 (GK1.1 clone) and α-CD8 (YTS clone) antibodies administered at 250 μg/dose were purchased from BioXcell. Human α-4-1BB antibody (BMS663513) was kindly provided by Bristol Meyer-Squibb. Human α-OX-40 antibody was unable to be obtained. Antibodies were diluted in culture media for in vitro studies and phosphate-buffered saline for in vivo studies.

**Thymidine incorporation cell proliferation assay**. Using spleens obtained from wild-type C57BL/6 mice, the Miltenyi MACS separation column system was used to isolate purified naive CD4+ and CD8+ T cells. T cells were seeded at $1 \times 10^5$ cells per well in 96-well plates and cells were either left unstimulated (no α-CD3/CD28) or activated with plate-bound α-CD3 (1 μg/ml) and soluble α-CD28 (0.5 μg/ml) antibody and treated with vehicle (0.1% DMSO), 2A3 isotype antibody (50 μg/ml), either α-4-1BB (50 μg/ml) or α-OX-40 (50 μg/ml) antibody alone, 100 nM of trametinib (MEKi) alone or combination of trametinib and α-4-1BB or α-OX-40 antibody, for an incubation period of 72 h. At 48 h, cells were pulsed with thymidine (0.5 μCi/well) and read at 72 h using the Tricarb 2910 TR liquid scintillation analyzer (Perkin Elmer).

**CFSE proliferation assay**. Human PBMCs were isolated and a Pan T-cell negative selection/enrichment (Miltenyi Biotec, San Diego, CA) was undertaken. T cells were rested overnight in RPMI and 10% FBS. The following day, cells were stained with CFSE per the manufacturer's protocol (Thermo Fisher), and $1 \times 10^5$ cells were seeded in 96-well plates containing α-CD3/α-CD28 (1:50) magnetic beads (Thermo Fisher), with or without trametinib (100 nM). After 96 h, cells were stained with α-CD4-APC (clone OKT4, Biolegend; 500 ng/ml) and α-CD8α-PE (clone HIT8α, Biolegend; 300 ng/ml). Population doubling for CD4+ and CD8+ T cells was determined by integrating CFSE+ peaks using stained, unstimulated cells as a control. Analysis was performed using the FlowJo software.

**Cytometric bead array**. CBA was performed utilizing supernatants obtained from in vitro mouse and human studies, BD capture beads for IFNγ were incubated for 1 h at room temperature, followed by a 1 h room temperature incubation of the PE detection beads. Following this, samples were analyzed using the FACS Verse, where output was displayed as pg/ml, as determined from the standard curve for each bead.

**T cell and tumor co-culture studies.** Spleens were isolated from OT-I (CD8[+]) and OT-II (CD4[+]) transgenic mice (obtained from the Walter and Elizabeth Hall Institute of Medical Research, Australia). Splenocytes were stimulated with either SIINFEKL (OT-I) or OVA$_{323-339}$ (OT-II) peptide (GenScript; 300 nM) for 3–4 days and supplemented with daily IL-2 (100 units/ml). AT3ova cells were either treated with 10–100 nM of trametinib for 12 h prior to co-culture (pre-treated 12 h MEKi) or trametinib was added directly to the co-culture for 24 h without prior pre-treatment (in culture 24 h MEKi). OT-I and OT-II cells were then co-cultured 1:1 with AT3ova cells ($1 \times 10^5$) for 24 h in the absence or presence of trametinib. Cells were collected and analyzed by FACS for expression of MHC-I, 4-1BB, and OX-40. Supernatants were collected for IFNγ (pg/ml) cytokine production by CBA.

**Mouse and human T-cell CBA analysis.** CD8[+] or CD4[+] mouse T cells were isolated from naive C57BL/6 spleens or human PBMCs (obtained from the Australian Red Cross Blood Service donation with informed consent; Agreement Number: 14-09VIC-04, NBMS Code:34ZPHR 003PHR) were isolated and purified for naive CD8[+]CD45[+] T cells using the MACS Miltenyi magnetic bead system. T cells were left unstimulated or activated with plate-bound α-CD3 (1 μg/ml) for 16 h (pre-stimulation). Cells were then washed and treated with vehicle (0.1% DMSO), α-CD28 (0.5 μg/ml in mouse only), 100 nM of MEKi alone, 10 μM of p38i or 10 μM JNKi alone, α-OX-40 (50 μg/ml) or α-4-1BB (50 μg/ml) antibody alone, or double or triple combination of trametinib, P38i, or JNKi and agonist antibody for 72 h (mouse) or 30 min—4 h (human). Following this, supernatants were collected and IFNγ cytokine production was measured via CBA analysis.

**Western blot analysis.** For western blot analysis, CD8[+] T cells were isolated from murine spleen or human PBMCs. About $1 \times 10^6$ T cells per condition were activated for 16 h with α-CD3 (1 μg/ml) prior to stimulation with α-CD28 (0.5 μg/ml) and the addition of MEKi (100 nM) and immune agonist (50 μg/ml) as described above in CBA analysis. Pellets were lysed after 72 h (mouse) or 5 min (human) using radioimmunoprecipitation assay buffer. Primary antibodies used for immunoblotting were purchased from Cell Signaling Technologies (phosphorylated); p-ERK (Thr202/Tyr204; #9101, 1:1000, 42/44 kDa), p-p38 (Thr180/Tyr182; D3F9 #9210, 1:1000, 43 kDa), p-MKK3/6 (Ser189/Ser207; D8E9 #12280, 1:1000, 38/40 kDa), p-MKK4 (Ser257; C36C11 #4514, 1:1000, 44 kDa), p-AKT (Ser473; D9E #4060, 1:1000, 60 kDa), p-JNK (Thr183/Tyr185; 81E11 #4668, 1:1000, 54 kDa), c-JUN (Ser63; 54B3 #2361, 1:1000, 48 kDa), p-NFκB (Ser536; 93H1 #3033, 1:1000, 65 kDa), and GAPDH (Abcam; ab-9484, 1:10,000, 40 kDa) loading control. Secondary antibodies used were α-rabbit (Santa Cruz; sc-2005, 1:2000) and α-mouse HRP IgG (Santa Cruz; sc-2030, 1:2000) for chemiluminescent signal detection. Band quantitation (densitometry) was undertaken using ImageJ (NIH). Briefly, areas were selected and histogram was produced by the software for band intensity displayed as an area value using the "Blot Quant" plugin. After highlighting peaks, the Analyze>Gels>Label Peaks function was used to express peaks as a percentage of the total size of all of the highlighted peaks. Values were then normalized to the untreated control for each antibody. Original blots are provided in the Supplementary Information.

**Genomic analysis of human breast cancer samples.** The expression levels (log2) and prognostic value of the MEK gene signature[30] as well as OX-40 and 4-1BB gene expression and survival data were analyzed from the METABRIC (Molecular Taxonomy of Breast Cancer International Consortium) data set[29]. Patient specimens were obtained with appropriate informed consent from the relevant institutional review board and committee. Access was granted and normalized data were downloaded from the European Genome Phenome Archive, extracted and analyzed in R. Clinical data for METABRIC were downloaded from cBioPortal[55, 56]. TNBC (Basal) samples were classified by the PAM50 molecular subtyper as previously published[55, 56]. Kruskal–Wallis tests were performed to compare expression levels across breast cancer subtypes. Kaplan–Meier survival curves were generated using tertiles of gene expression and differences tested using a Cox regression analysis using the gene as a continuous variable. Survival analyses were censored at 10 years. The end point used was relapse-free survival. We have evaluated and quantified TILs using our pre-defined method on 460 patients and 702 haematoxylin and eosin slides from TNBC patients downloaded from The Cancer Genome Atlas portal[32]. Correlations between the TILs and gene expression of OX-40, 4-1BB, and key immune genes in the TCGA data set were calculated using the Pearson correlation coefficient in R. All analyses were performed using R version 3.2.3.

**Genomic analysis of mouse samples.** RNA was extracted from 4T1ch9 tumors treated in vivo with vehicle (PEG 400/solutol) or trametinib (1 mg/kg/daily) for 7 days and Affymetrix Microarray was undertaken using the Affymetrix Mouse 430PM Array. Pre-processing and quantile normalization of microarray data was performed with the AFFY[57] package in R. From normalized RNA intensities, unbiased, differential expression analysis and a short list of genes (FDR < 0.05) was generated with the LIMMA[58] package in R. Using the differentially expressed genes, pathway and molecular function enrichment analysis was performed in MetaCore GeneGo[35]. The GSEA software[34] for gene set enrichment analysis was used with normalized RNA intensities. GSEA was used to determine the baseline

levels of a MEK signature (previously published by Pratilas et al.[30]) in AT3ova and 4T1Ch9 ex vivo tumors. False discovery rate was set at < 10%. Top genes were selected based on a P-value of < 0.05 and a fold change of > 1.5. All analyses were performed using R version 3.2.3. Data available from GEO under the accession code GSE101093.

**Sequencing of mutations and copy number analysis.** Whole-exome sequencing was performed on the AT3ova and 4T1Ch9 murine cell lines. Exome capture was performed using the Agilent SureSelectXT Mouse All Exon, and libraries passing QC were sequenced on an Illumina HiSeq 4000 to a mean fold coverage of ×160. Following alignment to the mouse reference genome GRCm38 with BWA[59], variants were called with GATK UnifiedGenotyper[60], Varscan2[60], and Platypus[61]. Only variants passing filters and called by Platypus and at least one other variant caller were analyzed. Variants were annotated with the Ensembl Variant Effect Predictor[62]. As the tumor model has been developed in several mouse backgrounds, there is no true matched normal sample, and the following strategy was employed to remove germline variants: (1) Removal of all known germline mouse variants using the current single-nucleotide polymorphism and indel calls provided by the Mouse Genomes Project[63]. To avoid missing germline variants due to inconsistent representation between call sets, RTG Tools "vcfeval" was used (Real Time Genomics, Hamilton, New Zealand). (2) Removal of variants found to occur in the Ensembl variation databases. The data have been deposited in the NCBI SRA under the accession code SRP103420.

**In vivo mouse studies.** Female C57BL/6 and Balb/c wild-type mice aged between 6 and 8 weeks were utilized. All experiments were conducted in accordance with the approval of the Peter MacCallum AEEC (Ethics number: E539). Treatment groups consisted of $n = 5$–8 mice per group. For in vivo experiments, using the C57BL/6 AT3ova model, $5 \times 10^5$ cells were resuspended in phosphate-buffered saline and injected as single cell suspensions, subcutaneously in a 100 μl volume into the right flank. For in vivo experiments using the Balb/c 4T1Ch9 model, $5 \times 10^4$ cells in a 20 μl volume of phosphate-buffered saline were injected into the fourth mammary fat pad. Treatments began on day 14 post inoculation for the AT3ova model, and day 10 for the 4T1ch9 model, with tumor sizes ranging between 35 and 60 mm$^3$. For experiments in RAG$^{-/-}$ mice, AT3ova model, $5 \times 10^5$ cells were resuspended in phosphate-buffered saline and injected as single cell suspensions subcutaneously in a 100 μl volume into the right flank. Tumor volumes were calculated using the equation (length × width$^2$)/2, where length and width refer to the larger and smaller dimensions collected at each measurement. Following the establishment of tumors, mice were treated with vehicle control (suspension agent), trametinib (1 mg/kg orally, once daily), α-4-1BB alone (25 μg/dose delivered intraperitoneally), or α-OX-40 antibody alone (50 μg/dose in Balb/c and 200 μg/dose in C57BL/6 by intraperitoneal injection) or isotype control antibody, or double combination of trametinib and agonist immunotherapy (α-4-1BB Ab/α-OX-40 Ab). For experiments using the triple combination, α-PD-1 antibody was administered in double combination with trametinib, or either agonist or in triple combination with trametinib, α-PD-1 Ab (100 μg/dose in Balb/c and 200 μg/dose in C57BL/6 by intraperitoneal injection), and either α-4-1BB Ab/α-OX-40 Ab. For depletion experiments in the wild-type C57BL/6 AT3ova model, α-CD4 (250 μg/dose) and α-CD8 (250 μg/dose) antibodies were administered concurrently on day −1, 0, 7, and 14. Immunotherapy and isotype controls were delivered on days 0, 4, 8, and 12 in C57BL/6 mice and 0, 5, 10 in Balb/c mice. Tumor volume was measured 2–3 times weekly with calipers. Survival was monitored and determined when the tumors reached an ethical limit of 1400 mm$^3$.

**FACS analysis.** For ex vivo studies, tumors were dissected, mechanically minced, and digested using a mixture of 1 mg/ml collagenase type IV (Sigma-Aldrich) and 20 μg/ml DNAase (Sigma-Aldrich). After 30 min of digestion at 37 °C, the cells were passed through a 70 μm filter, washed and then passed through a 40 μm filter. Single cell suspensions were then stained with antibody cocktails for various TIL subsets. In some experiments, isolated cells were restimulated with phorbol 12-myristate 13-acetate (PMA) (50 ng/ml) and ionomycin (1 μg/ml; Sigma-Aldrich) in the presence of GolgiPlug (BD Biosciences; 1:1000) and GolgiStop (BD Biosciences; 1:1500) for 4 h prior to flow cytometry analysis. Samples were analyzed by FACS with Fixable Yellow used to discriminate viable and dead cells. Lymphocytes were distinguished by CD3[+]/TCRβ[+], CD45[+] cells. BD fluorosphere counting beads were added to cocktails before running samples. A total of 20,860 beads were added per sample.

**Statistical analysis.** Statistical analyses were performed with Prism, version 6 (GraphPad Software Inc.). All data are presented as mean ± SEM. Means for all data were compared by unpaired Student's t-test or one-way ANOVA for in vitro studies and two-way ANOVA with post hoc Fisher's least significant difference testing for in vivo studies. Log ranked (Mantel–Cox) analysis was undertaken for survival proportions. P-values of <0.05 were considered statistically significant. *$P < 0.05$, **$P < 0.01$, ***$P < 0.001$, ****$P < 0.0001$.

**Data availability**. The Mouse tumor whole-exome sequencing data have been deposited into the National Center for Biotechnology Information Sequence Read Archive under the accession number SRP103420. The Mouse Affymetrix Microarray data referenced during the study are available in a public repository from the Gene expression omnibus (GEO) website under the accession code GSE101093; http://www.ncbi.nlm.nih.gov/geo/query/acc.cgi?acc=GSE101093. The human sample data referenced from TCGA (available at: https://tcga-data.nci.nih.gov/) and METABRIC (available at: http://www.cbioportal.org/study?id=brca_metabric#summary) are open access data sets that are publicly available. The authors declare that all the other data supporting the findings of this study are available with the article and its supplementary information files, and from the corresponding authors upon reasonable request.

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

## Acknowledgements

We thank Ralph Rossi, Vicki Milovac, and Sophie CurcioPeter from MacCallum Cancer Centre (PMCC) FACS facility. Kaushalya Amarasinghe; PMCC Animal facility and technicians, PMCC Cytotoxic suite, PMCC Bioinformatics core. Laura Kilpatrick and Amy Rogers (Cancer Cell Death Laboratory) for providing the OT-I cells. This work was funded by project grants from the National Breast Cancer Foundation (NBCF), Australia and the National Health and Medical Research Council (NHMRC). S.L. is supported by Cancer Council, Victoria, Australia and the Breast Cancer Foundation (BCRF), N.Y. P.A.B. is supported by Fellowships (ID# PF-14-008). P.K.D. and M.H.K. are supported by NHMRC Senior Research Fellowships (APP1041828 and APP1058388), respectively.

## Author contributions

Conception and design: S.L., P.K.D., P.A.B., S.D. Development of methodology: S.L., P.K.D., P.A.B., S.D. Acquisition of data (provided animals, acquired, and managed patients, provided facilities, etc.): S.L., P.K.D., P.A.B., S.D., F.C., J.M.B. Analysis and interpretation of data (e.g., statistical analysis, biostatistics, computational analysis): S.L., P.K.D., P.A.B., S.D., F.C., J.M.B. Writing, review, and/or revision of the manuscript: S.D., Z.L.T., F.C., B.V., C.M., M.A.H., P.S., S.J.L., M.M., M.H.K., J.A.T., P.J.N., R.S., G.A.M., J.M.B., P.A.B., P.K.D., S.L. Administrative, technical, or material support (i.e., reporting or organizing data, constructing databases): S.L., P.K.D., P.A.B., S.D. Study supervision: S.L., P.K.D., P.A.B.

## Additional information

**Competing interests:** S.L.'s lab receives research funding from Novartis. The remaining authors declare no competing interests.

