## [Peer Review File · Nature Communications]

Reviewers' comments:

Reviewer #1 (Remarks to the Author):

This manuscript builds on the recently published observation by the same investigators that TNBC tumors with Ras/MAPK activation have decreased levels of TILs and poor prognosis, suggesting that this pathway is involved in regulating tumor immunogenicity. Here, the authors hypothesize that MEK inhibitors could restore tumor immunogenicity. Moreover, they hypothesize that the therapeutic effectiveness of MEKi is partially hindered by the fact that the same pathway is needed for T cell activation, proliferation, function and survival. Thus, they test the hypothesis that interventions that can overcome the negative effects of MEKi tramatenib on T cells could be used in combination with tramatenib to achieve a synergistic anti-tumor effect. They choose to test anti-41BB and anti-OX40 antibodies because these costimulatory pathways have been shown to work via an alternative signaling pathway not requiring MEK. The latter point is confirmed in *in vitro* experiments.

The hypothesis is intriguing and clinically relevant, and obtained results overall support the concept that a combination of MEKi with either anti-41BB or anti-OX40 could be effective in TNBC and possibly other cancers. However, novelty is limited since the same investigators recently showed that the tumor responses to trametinib was improved by anti-PDL-1 (ref 15). In addition, data shown do not provide convincing evidence in support of the main conclusions that (1) tramatenib increases tumor immunogenicity *in vivo* and its effect is partially mediated by T cells; (2) the improved tumor control in mice treated with tramatenib+anti-41BB or tramatenib+anti-OX40 is due to an interaction of these agents that leads to recovery of the signaling directly affected by MEKi. In fact, the effects of anti-41BB and anti-OX40 are very similar to the effects of anti-PD-L1 (ref 15). Importantly, it is not clear that the mouse tumor models used are modeling the human TNBC with oncogenic activation of MAPK pathway, which have poor TIL infiltrate. AT3-OVA is likely to be relatively immunogenic since OVA is a strong exogenous antigen, and 4T1ch9 is tagged with Cherry. The parental 4T1 line is known to be poorly immunogenic and highly metastatic. However, metastases are not analyzed here and the reason to use a derivative expressing an antigenic fluorescent protein is not explained. Overall, the manuscript lacks mechanistic depth. Other specific points are listed below.

1) The increased immunogenicity of cancer cells exposed to tramatenib is only shown by *in vitro* treatment of AT3ova cells, which results in increased MHC-I expression (Fig 2). This is not novel data since the same authors have recently published that MHC and PDL-1 were upregulated by MEKi in the same mouse tumors (ref 15). Many other receptors (NKG2D ligands, Fas, etc..) and soluble factors (e.g., chemokines) should be tested *in vitro*, and cells analyzed for phenotypic changes also after *in vivo* treatment. In Figure 7 tumors are analyzed for gene expression changes after treatment of mice with trametinib. However, data are not too convincing since there is no clear Th1 gene signature and the untreated tumors may be significantly bigger (i.e., more hypoxic and immunosuppressed) than the treated tumors but a more comprehensive analysis of their composition is not performed. Experiments to demonstrate the role of T cells in tumor control achieved with tramatenib alone (e.g., T cell depletion) should be performed.

2) In Figure 2, the analysis of T cell density is expressed as percentage of CD45+ cells. Thus, the apparent decrease in T cells may simply reflect an increase in myeloid cells. The decreased function of T cells could also be an indirect effect of increased tumor infiltration by MDSC in treated tumors. In fact, an increase in myeloid cells has been described after other types of treatments, including chemotherapy. Although there is some evidence that trametinib decreases MDSC (ref 42), MDSC and TAM should be analyzed in this study.

In addition, since over the time of observation (day 2 to 9) treated and untreated tumors do not grow the same way, instead of showing data as a fold change the actual density of T cells should be shown in both groups. The apparent "recovery" in T cell percentage seen at day 9 in treated animals could instead represent a loss of T cells in untreated tumors, an explanation that likely

accounts for the unexplained fact that T cells become resistant to MEKi overtime.

3) Figure 4A; it is not clear what the numbers indicate, and the CFSE dilution results are difficult to understand as presented. Also, the reason for the analysis at 30, 60 and 120 minutes in B is not clear.

4) Figure 5: Why mice with 4T1Ch9 tumors are treated only for 15 days with trametinib? There is a small survival advantage in this model with anti-OX40 alone (Fig 5H) despite the lack of any effect on tumor growth (Figure 5D): do mice die of lung metastases or are they euthanized for reaching a pre-determined primary tumor size?

5) Figure 6 and suppl fig 2-3: The immune infiltrate is characterized in the tumors but there is no measure of tumor-specific T cells. The response to OVA can be easily measured, and there are several known endogenous antigens in 4T1, thus expansion of antigen-specific T cells can and should be assessed.

Reviewer #2 (Remarks to the Author):

General comments

The work presented here builds on demonstration and other contexts that in MAP kinase pathway prevent tumors, MEK inhibition is associated with increased T cell and filtration. And, others have shown that MEK inhibitors suppress interferon production and proliferation following antigen exposure. There is novelty in the finding that 4-1BB and OX40 agonists can reverse this effect.

Data mining performed in support of a role for the MAP kinase pathway in a subset of TNBC and its relationship with 4-1BB and OX-40 expression provides nice support for the experimental work to follow.

The demonstration that p38 and JNK inhibitors reverse the positive effects of 4-1BB and OX-40 antibodies is an intriguing aspect of this series of experiments.

The glaring absence of this work is use of a PD-1 or PD-L1 antibody alone and in combination with a MEK inhibitor in order to contextualize the observed effects of 4-1BB and OX-40 antibodies. While the authors correctly note that there has been modest single connectivity with PD-1/PD-L1 antibodies in TNBC, it is entirely possible that MEK would substantially augment their activity. And, while PD-1/PD-L1 antibodies have a modest response rate in TNBC, they are still more active than 4-1BB and OX-40 antibodies as single agents. In the end, a MEK/PD-1 combination strategy would be of immediate translational relevance.

Additionally, the experiments focus on tumor cell and T cell interactions ex vivo and in vivo with little consideration of the effect of an examination on constituents of the tumor microenvironment (tumor associated macrophages and MDSC, in particular, given the number of papers published on the topic of the importance of MAPK signaling in the cells).

While the use of two immune competent walls of breast cancer is a strength, the observed combined efficacy of an examination with either 4-1BB or OX-40 antibodies is a disappointment in that slowing of growth is observed rather than regressions or clearance of tumor.

Specific comments

Introduction, "The complex interplay between the kinetics of MEK inhibition (MEKi) on T cell function and its relevance to the therapeutic efficacy of MEKi in solid cancers is currently undefined."

The author should acknowledge the work of Ebert PJ et al. MAP Kinase Inhibition Promotes T Cell and Anti-tumor Activity in Combination with PD-L1 Checkpoint Blockade. *Immunity*. 2016 Mar 15;44(3):609-21. as it is directly relevant to the area they explore.

The experiments performed in support of the effect of MEK inhibitors on TNBC tumor cell immunogenicity suffer from a lack of specific focus on tumor cells alone (by flow or in situ).

Discussion, "Although the evidence suggests that targeting MEK may increase tumor immunogenicity, potentially allowing for greater TIL infiltrate, MEKi has long been speculated to adversely affect anti-tumor immunity"

This has been more than a pointless speculation. Several groups of directly demonstrated the effect of MEK inhibitors on effector T cell function.

Discussion, the brief mention of the Ebert et al and Boni et al papers downplays their contribution in this space. This work is sufficiently foundational that it warrants mention in both the introduction and discussion.

Reviewer #3 (Remarks to the Author):

The presence of tumor infiltrating lymphocytes (TIL) is a positive prognostic factor in several tumor types including triple negative breast cancer (TNBC). Recent studies have demonstrated that immunotherapies can enhance TIL function and there is evidence of clinical activity of checkpoint blockade in women with TNBC. However, the mechanisms regulating TIL recruitment and function within TNBC remain poorly understood. Previous studies have demonstrated that increased Ras/MAPK pathway activity correlates with reduced levels of TIL, implicating this signaling axis as a potential means of modulating the tumor microenvironment (TME) in TNBC. Since this pathway also regulates T cell function and survival, it is critical to understand the potential negative effects that MEK inhibition (MEKi) can have on immune infiltrates within the TME. Previous work has shown that T cell agonist mAb therapy including anti-4-1BB (a4-1BB) and anti-OX40 (aOX40) stimulate T cell responses in a MEK-independent manner. Based upon these data, the authors hypothesized that the combination of MEKi plus T cell agonist immunotherapy could overcome the potential inhibitory effects of MEKi on T cells to enhance anti-tumor immunity. They demonstrate that T cell agonist immunotherapy in conjunction with MEKi restored T cell proliferation and function in the TME, leading to improved tumor control in a pre-clinical model of TNBC.

Overall, this is a very insightful study that provides insight into the mechanisms by which a4-1BB and aOX40 immunotherapy can enhance T cell function in the presence of MEKi. The inclusion of relevant human and murine data strengthens the authors' conclusions that the indicated immunotherapeutic agents can restore the function of TIL in vivo. However, there are several concerns that should be addressed.

- 1) In Figure 1C and 1D, it would be helpful to determine whether the association of increased 4-1BB or OX40 gene expression with improved survival merely reflects differences in T cell infiltrates. This should be addressed and discussed in the results.
- 2) Figure 2 shows effects upon IFN- γ production when MEK is given in culture for 24 hr. Given the effects of the drug on cell cycle progression, are these effects due to impaired T cell

activation/proliferation? Specifically, does the lack of cytokine production reflect a general impairment in TCR stimulation, rather than a more specific impairment in IFN-g secretion? If the cells were unable to proliferate, then one would not expect them to differentiate and acquire effector function.

3) Also related to Figure 2, is there any evidence of increased MHC I expression (by IHC or flow cytometry) following MEKi treated in vivo or is this only detected in vitro?

4) In Figure 2, it is difficult to determine the effects of the treatment on T cell frequencies and/or function given how the data is presented. Total frequencies (% and total number) would be helpful, rather than fold-changes in expression.

5) The proliferation data present in Figure 3 are intriguing, however it is not clear whether the addition of immune agonists has activity in this assay as there is little or no enhancement in proliferation with these agents alone. Also, the addition of immunotherapy plus MEKi appears to restore proliferation back to baseline levels, which bring up the issue as to what is happening mechanistically – are the cells somehow protected from the proliferative block in the presence of these other signals? This could potentially be a somewhat different interpretation than the use of an alternative signaling pathway. It would be useful to evaluate the effects of MEKi plus aOX40 or a4-1BB therapy in an in vivo priming model using OT-I T cells to elucidate the effects of this combination in a setting where a robust co-stimulatory signal is provided.

6) The data presented in Figure 4B is perplexing as there is no difference in IFN-g levels at 240 min post-treatment suggesting a very transient effect with MEKi. It isn't clear why there isn't a sustained effect on the MEKi monotherapy group. This should be discussed.

7) The authors should discuss the tumor growth/survival data in Figure 5 – it is intriguing that none of the treatments led to complete tumor regression. Is this related to the MEKi treatment or just these particular tumor models? Have any other models been tested to see if durable CRs can be achieved?

8) The references (Refs. 26, 27) provided as part of the rationale for examining the role of aOX40/a4-1BB and MEK-independent activation/signaling appear to be incorrect and/or missing. Please update/correct as needed. Also, recent work from the Ribas group (OncoImmunology 5:7, e1052212; July 2016) should be discussed as they also examined the effects of BRAF inhibitors plus a4-1BB mAb therapy.

9) Representative histograms and/or dot plots should be provided for the associated flow cytometry data presented in Figures 2 and 6.

We thank all the reviewers for their time and comments on our manuscript. We believe that our responses to these comments have significantly improved the manuscript.

Response to Reviewer 1 comments:

1) It is not clear that the mouse tumor models used are modelling the human TNBC with oncogenic activation of MAPK pathway, which have poor TIL infiltrate.

Author's response:

The purpose of using these two cell lines is that the AT3ova emulates a high TILs environment while the 4T1Ch9 represents a low TILs environment, allowing us to model both 'hot' and 'cold' inflamed tumors in the patient setting. Our previously published data on these cell lines show clear MEK 1/2 activation and expression at the proteomic level (Loi *et al*, 2016; Clin Can Res).

2) The reason to use a derivative expressing an antigenic fluorescent protein is not explained.

Author's response:

The 4T1Ch9 line has similar growth kinetics when compared with the parental 4T1 cell line, indicating that it is not immunogenic. This is consistent with other reports showing that cherry fluorescent protein is not immunogenic in the BALB/c background (Eckhardt *et al*, 2005; Mol Can Res). Additionally, we can use the cherry to detect tumor cells via FACS analysis.

3) The increased immunogenicity of cancer cells exposed to trametenib is only shown by in vitro treatment of AT3ova cells, which results in increased MHC-I expression (Fig 2). Many other receptors (NKG2D ligands, Fas, etc..) and soluble factors (e.g., chemokines) should be tested in vitro, and cells analyzed for phenotypic changes also after in vivo treatment.

Author's response:

We have previously shown that MEKi increases MHC-I, MHC-II and PDL-1 on both AT3ova and 4T1Ch9 tumor cells *in vivo* (Loi *et al*, 2016; Clin Can Res). We now present new data showing the expression of other receptors/ ligands following MEK inhibition. This experiment revealed that Fas, TRAIL and NKG2DL (RAE-1) were significantly upregulated in the presence of MEKi *in vitro* (New Figure 2A, B). We also observed a trend for increased expression of these receptors *in vivo* following MEK inhibition. This data is shown in Supplementary Figure 1A and is referred to in the following text in the results section on page 5 of the tracked manuscript:

"We have previously shown that MEKi increases MHC-I, MHC-II and PDL-1 on both AT3ova and 4T1Ch9 tumor cells in vivo¹⁵. To further characterise the effect of MEKi on tumor immunogenicity we examined the expression of other receptors and ligands. We observed that the expression of Fas, TRAIL and NKG2D (RAE-1) were significantly upregulated in the presence of MEKi in vitro in both the AT3ova and 4T1Ch9 cell lines (Figure 2 A, B). We also observed a trend for increased expression of FAS, TRAIL and NKG2D on AT3ova tumor cells in vivo (Supplementary Figure 1A). Given that the most pronounced effects of MEKi were on MHC-I expression (Figure 2 A, B), we next explored the effects of MEKi induced MHC-I expression on tumor cells and subsequent T cell responses. "

In terms of chemokines, our data in Figure 7A (Now Figure 10A in the revised manuscript) shows that that in MEKi treated 4T1Ch9 tumors several chemokines/ chemokine receptors were modulated at the transcriptional level. This data was referred to in the following text in the Results section located on page 10 of the revised manuscript.

"We found that immune genes such as CD3 ϵ , CD8 α , CD28, TCR- β , TNFs and various chemokine receptors (CXCR5, CCR9, CCL19-21) were upregulated in the MEKi treated tumors (Figure 10A) consistent with the rebound in TIL infiltrate observed at day 7."

To expand on this point, we have now also referred to this data in the Discussion section on page 15 of the revised manuscript.

“Many of these chemokines and cytokines have been implicated in the immune response in breast cancer. For example, some breast cancer cell lines have been shown to express CXCR4-CXCL12, CCR6. CCR9 and CCL20, which are involved in promoting metastasis by enhancing tumor cell proliferation and migration⁴⁷. Additionally, CCL19 has been found to activate T-cells and CCL21, CXCR5-CXCL13 has been shown to regulate naive T-cell homing to secondary lymphoid organs⁴⁸.”

4) In Figure 7 tumors are analyzed for gene expression changes after treatment of mice with trametinib. However, data are not too convincing since there is no clear Th1 gene signature.

Authors response:

We have included additional data in Figure 7A (Now Figure 10A in the revised manuscript) which highlights the TH1 response, including gene expression for TH1 associated genes such as TNF, CCL5, CXCR5, CXCR3, Tbx21, Eomes, Icos, Stat4, Nfatc, Il27ra, Tnfsf4, Cd40, Prf1 (Perforin), which were significantly upregulated. This new data is referred to in the following text on page 10 of the Results section of the revised manuscript.

“Additionally, we observed many genes associated with a Th1 signature (TNF, CCL5, CXCR5, CXCR3, Tbx21, Eomes, Icos, Stat4, Nfatc, Il27ra, Tnfsf4, Cd40, Prf1) that were significantly upregulated following MEKi”.

5) Experiments to demonstrate the role of T cells in tumor control achieved with trametinib alone (e.g., T cell depletion) should be performed.

Authors response:

To address the role of T cells in the response to trametinib alone we have performed additional experiments treating AT3ova tumor bearing mice on the RAG^{-/-} background (New Figure 9A-B) and in T cell depleted wild type mice (New Figure 9C-D). These experiments revealed that the absence of T cells enhanced the growth of tumors of non-treated mice but did not affect the efficacy of MEKi. This is consistent with our data indicating that MEKi inhibits the effectiveness of anti-tumor T cell responses by suppressing T cell effector functions. Our experiments also revealed that, as expected, the addition of immune agonists (anti-4-1BB or anti-OX-40) enhanced anti-tumor effects in combination with MEKi in T cell replete but not in T cell depleted mice. This new data is referred to in the following text in the Results on page 10 and in the Discussion on page 12 of the revised manuscript.

“To demonstrate the importance of T cells in the therapeutic effects observed with MEKi and anti-OX-40/ anti-4-1BB, we undertook further studies in RAG^{-/-} mice (which lack T and B cells) and in WT mice depleted of CD4⁺/CD8⁺ T cells. We show that in both RAG^{-/-} (Figure 9A,B) and CD4⁺/CD8⁺ T cell depleted WT mice (Figure 9C,D), that the addition of agonist immunotherapies (α-4-1BB and α-OX-40) had no additional effect to MEKi alone (Figure 9A-D).”

“To support this we show that the anti-tumor efficacy of MEKi is similar in T cell depleted (either RAG^{-/-} or antibody depleted) or non-depleted mice. This is consistent with our data indicating that MEKi inhibits the effectiveness of anti-tumor T cell responses by suppressing T cell effector functions. Our experiments also revealed that, as expected, the addition of immune agonists (anti-4-1BB or anti-OX-40) enhanced anti-tumor effects in combination with MEKi in T cell replete but not in T cell depleted mice.”

6) In Figure 2, the analysis of T cell density is expressed as percentage of CD45+ cells. Thus, the apparent decrease in T cells may simply reflect an increase in myeloid cells.

Author's response:

To confirm that observed effects of MEKi on T cell frequencies was not due to modulation of other immune subsets we performed an additional experiment to quantify absolute numbers of immune cell subsets infiltrating tumors. These experiments revealed that only CD8⁺ and CD4⁺ T cell numbers were reduced following MEKi, whilst the numbers of other immune subsets including macrophages and

MDSCs remained constant (new Supplementary Figure 4B). This new data is referred to on page 7 and 9 of the Results section and page 14 of the Discussion section of the revised manuscript.

“To confirm that the observed effects of MEKi on T cell frequencies was not due to modulation of the frequency of other immune subsets we next quantified absolute numbers of various immune cell subsets infiltrating tumors. These experiments revealed that the number of CD45⁺ cells remained constant in the vehicle and MEKi treated groups at day 4 (Supplementary Figure 4A). Furthermore, this analysis showed that only CD8⁺ and CD4⁺ T cell numbers were reduced following MEKi (Supplementary Figure 4B), whilst the numbers of other immunosuppressive subsets including macrophages (CD11b⁺ F4/80⁺ TAMs) and MDSCs (CD11b⁺ Ly6C⁺/Ly6G⁺) remained constant (Supplementary Figure 4B). Both CD4⁺ FOXP3⁻ T cells and CD4⁺ FOXP3⁺ Tregs showed an overall decrease in cell numbers (Supplementary Figure 4C). Analysis of tumor specific T cells using the H2Kb OVA (SIINFEKL) tetramer, revealed that MEKi similarly reduced the number of both tumor antigen specific CD8⁺ T cells (tetramer positive) and CD8⁺ T cells recognising unknown antigens (tetramer negative) (Supplementary Figure 4C). This indicates that the MEKi induced inhibition is a global effect across all CD8⁺ T cells.”

“While several studies have shown increased infiltration of TAMs and MDSCs following treatment^{21, 22, 39}, we show that these populations were unchanged following MEKi treatment alone. This is potentially due to the different effects of MEKi and the BRAFi used in these studies. Interestingly, in our study MEKi reduced the frequency of TAMs, MDSCs and Treg subsets when combined with either anti-OX-40 or anti-4-1BB antibody, potentially owing to the fact that these subsets are driven by MAPK signalling^{45 46}.”

7) Figure 4A; it is not clear what the numbers indicate, and the CSFE dilution results are difficult to understand as presented. Also, the reason for the analysis at 30, 60 and 120 minutes in B is not clear.

Author’s response:

The numbers indicate the number of divisions. We have amended the figure and figure legend (Now Figure 5 in the revised manuscript) to reflect this more clearly.

With regards to the reason for analysis at the specified timepoints, it is important to note that there are differences in the timing of signaling events between mouse and human T cells. The signaling events in human T cells occur very early on and as such the early time points were chosen in order to capture these rapid events. We have added the following text in the Discussion on page 14 of the revised manuscript to clarify this point.

“Interestingly, our studies in mouse and human T cells highlight some differences in the duration of signaling events, as well as some potential compensation in the pathways. This redundancy between mouse and human systems has previously been discussed³⁹. Nonetheless, we observe the same inhibitory effect on T cell signaling following MEKi in both mouse and human T cells, thus validating these findings.”

8) Figure 5: Why mice with 4T1Ch9 tumors are treated only for 15 days with trametinib?

Author’s response:

The trametinib dosing schedule was designed to coincide with antibody administration. Further doses of antibodies are not possible in this model due to antibody mediated lethal toxicity observed in the BALB/c background. Please note that this data is now in Figure 6 of the revised manuscript.

9) There is a small survival advantage in this model with anti-OX-40 alone (Fig 5H) despite the lack of any effect on tumor growth (Figure 5D): do mice die of lung metastases or are they euthanized for reaching a pre-determined primary tumor size?

Author’s response:

The end point of the experiment represents the time at which tumors have reached their ethical size limit, rather than death. Whilst this information was originally stated in the Materials and Methods, we have now added this information to the Figure legend for clarity. Therefore, the small survival advantage represents a small inhibition of tumor growth by anti-OX-40, which results in tumors taking longer to reach 1400 mm³. We have adjusted the scale of the graph in Figure 6D of the revised manuscript to make this more clear.

10) Figure 6 and suppl fig 2-3: The immune infiltrate is characterized in the tumors but there is no measure of tumor-specific T cells.

Author's response:

To address this concern we have undertaken an additional experiment involving the use of the fluorescently tagged OVA Kb Tetramer in order to determine the role and function of CD8 tumor-antigen specific T cells. This new data shows that both antigen specific and non-specific cells are similarly affected by MEKi. This data is shown in new Supplementary Figure 4C and is referred to in the Results section on page 7 of the revised manuscript.

"Analysis of tumor specific T cells using the H2Kb OVA (SIINFEKL) tetramer, revealed that MEKi similarly reduced the number of both tumor antigen specific CD8⁺ T cells (tetramer positive) and CD8⁺T cells recognising unknown antigens (tetramer negative) (Supplementary Figure 4C)."

Response to Reviewer 2 comments:

1) The glaring absence of this work is use of a PD-1 or PD-L1 antibody alone and in combination with a MEK inhibitor in order to contextualize the observed effects of 4-1BB and OX-40 antibodies.

Author's response:

We have previously shown that anti-PD-1 can enhance the therapeutic effects of MEKi (Loi *et al.* 2016, CCR). To address this question we have performed further experiments to compare the efficacy of anti-PD-1 to the therapeutic effects observed with anti-4-1BB or anti-OX-40 in the context of MEK inhibition. Strikingly, in the 4T1Ch9 model, the efficacy of either anti-4-1BB/ MEKi or anti-OX-40/ MEKi was significantly greater than the combination of anti-PD-1 and MEKi (Figure 7A-D). This suggests that combining MEKi with anti-OX-40 or anti-4-1BB may be more effective than MEKi and anti-PD-1 antibody in treating 'cold' tumors with low TILs. Furthermore, since we observed increased expression of PD-1 on tumor-infiltrating CD8⁺ T cells following anti-4-1BB/ anti-OX-40 treatment (data not shown) we next investigated the efficacy of combining MEKi with both an agonist immunotherapy regimen (anti-4-1BB/ anti-OX-40) and blockade with α -PD-1. We found that the survival of mice was significantly improved in both tumor models (Figure 7C-F), particularly in the α -OX-40 combination, with 70% of AT3ova bearing mice surviving over 120 days (Figure 7F). This new data is referred to on page 8-9 of the revised manuscript.

"We have previously shown that anti-PD-1 can enhance the therapeutic effects of MEKi¹⁵ and so we compared the effects observed with anti-4-1BB and anti-OX-40. Strikingly, in the 4T1Ch9 model, the efficacy of either anti-4-1BB/ MEKi or anti-OX-40/ MEKi was significantly greater than the combination of anti-PD-1 and MEKi (Figure 7A,B). Furthermore, since we observed increased expression of PD-1 on tumor-infiltrating CD8⁺ T cells following anti-4-1BB/ anti-OX-40 treatment (data not shown) we next investigated the efficacy of combining MEKi with both an agonist immunotherapy regimen (anti-4-1BB/ anti-OX-40) and blockade with α -PD-1 antibody. We found that the survival of mice was significantly improved (Figure 7C-F), particularly in the α -OX-40 combination, with 70% of AT3ova bearing mice surviving over 120 days (Figure 7F)."

2) Additionally, the experiments focus on tumor cell and T cell interactions ex vivo and in vivo with little consideration of the effect of an examination on constituents of the tumor microenvironment (tumor associated macrophages and MDSC, in particular, given the number of papers published on the topic of the importance of MAPK signaling in the cells).

Author's response:

These points have been addressed in our response to Reviewer 1 (**Question 6**).

3) While the use of two immune competent walls of breast cancer is a strength, the observed combined efficacy of an examination with either 4-1BB or OX-40 antibodies is a disappointment in that slowing of growth is observed rather than regressions or clearance of tumor.

Author's response:

We would argue that regression and tumor clearance are rarely achieved in these models, and as such they underestimate the potential for these combinations in patients. However, our new experiments with the inclusion of anti-PD-1 (**see answer to Question 1 above**) result in dramatic increases in survival in the triple combinations in both the AT3ova and 4T1Ch9 models.

4) Introduction, "The complex interplay between the kinetics of MEK inhibition (MEKi) on T cell function and its relevance to the therapeutic efficacy of MEKi in solid cancers is currently undefined." The author should acknowledge the work of Ebert PJ et al. MAP Kinase Inhibition Promotes T Cell and Anti-tumor Activity in Combination with PD-L1 Checkpoint Blockade. Immunity. 2016 Mar 15;44(3):609-21. as it is directly relevant to the area they explore.

Author's response:

We have adjusted the text to further highlight the reference. Please refer to the Introduction section on page 3 of the revised manuscript where the following text has been included.

"Limited studies have undertaken in depth exploration into the effects of MEKi on T cells functionality, where most reports have been somewhat contradictory. Some studies have shown that MEKi potentiates anti tumor immunity^{23, 25}, while others suggest that MEKi only transiently inhibits T cell function^{21, 22}. As such, in this study we aimed to investigate the long-term effects of MEKi on T cells."

5) The experiments performed in support of the effect of MEK inhibitors on TNBC tumor cell immunogenicity suffer from a lack of specific focus on tumor cells alone (by flow or in situ).

Author's response:

These points have been addressed in the responses to reviewer 1 (**Question 3**).

6) Discussion, "Although the evidence suggests that targeting MEK may increase tumor immunogenicity, potentially allowing for greater TIL infiltrate, MEKi has long been speculated to adversely affect anti-tumor immunity". This has been more than a pointless speculation. Several groups of directly demonstrated the effect of MEK inhibitors on effector T cell function.

Author's response:

We have changed this text (located on page 12) of the revised manuscript to reflect this as follows.

"Although the evidence suggests that targeting MEK may increase tumor immunogenicity, potentially allowing for greater TIL infiltrate, MEKi has also been shown to adversely affect anti-tumor immunity".

7) Discussion, the brief mention of the Ebert et al and Boni et al papers downplays their contribution in this space. This work is sufficiently foundational that it warrants mention in both the introduction and discussion.

Authors response:

We have adjusted the text to reflect this. Please refer to page 3 of the Introduction in the revised manuscript.

“Limited studies have undertaken in depth exploration into the effects of MEKi on T cells functionality, where most reports have been somewhat contradictory. Some studies have shown that MEKi potentiates anti tumor immunity^{23, 25}, while others suggest that MEKi only transiently inhibits T cell function^{21, 22}. As such, in this study we aimed to investigate the long-term effects of MEKi on T cells.”

These papers are also mentioned on page 13-14 of the Discussion in the revised manuscript.

“This observation of T cell inhibition with MEKi is consistent with the findings of Ebert et al. who suggested that MEKi prevented T cell priming in lymph nodes²³. Similarly, Hu-Lieskovan and colleagues showed that cytokine production was decreased in vitro, although they observed that these effects were not as profound in vivo²². However our study convincingly demonstrates that these inconsistencies are likely explained by the time points at which ex-vivo tumors and TILs were analysed in other studies^{21, 22}. Both Boni et al. and Vella et al. found that MEKi alone, inhibited T cell proliferation, antigen-specific expansion and cytokine production^{24, 25}”.

Response to Reviewer 3 comments:

1) In Figure 1C and 1D, it would be helpful to determine whether the association of increased 4-1BB or OX-40 gene expression with improved survival merely reflects differences in T cell infiltrates. This should be addressed and discussed in the results.

Author’s response:

The increase in 4-1BB and OX-40 are reflective of an increase in TILs. We show this correlation in **Figure 1E**. To clarify this, we have included the following text on page 5 of the Results section of the revised manuscript.

“The strong positive correlation between TILs and 4-1BB/ OX-40 expression (Figure 1E), likely explains the association with 4-1BB/ OX-40 and improved patient outcomes (Figure 1C, D).”

2) Figure 2 shows effects upon IFN-g production when MEK is given in culture for 24 hr. Given the effects of the drug on cell cycle progression, are these effects due to impaired T cell activation/proliferation?

Author’s response:

Given that MEK signaling occurs downstream of TCR activation, we would expect that both proliferation and cytokine production are affected concurrently. To clarify this the following text has been inserted on page 12 of the Discussion section of the revised manuscript.

“Herein, we show in models of TNBC that MEKi has early detrimental effects on T cell function in terms of proliferation and cytokine production. Since MEK signalling occurs downstream of TCR activation (Supplementary Figure 10) it is likely that MEKi affects these parameters concurrently.”

3) Also related to Figure 2, is there any evidence of increased MHC I expression (by IHC or flow cytometry) following MEKi treated in vivo or is this only detected in vitro?

Author’s response:

We have previously published that MEKi increases MHC-I, MHC-II and PDL-1 expression *in vivo* in these tumor models (Loi et al., 2015; Clin Can Res). We have performed additional experiments, confirming this observation which is shown in Figure 2 of the revised manuscript. This is described in the following text on page 5 of the revised manuscript.

“We have previously shown that MEKi increases MHC-I, MHC-II and PDL-1 on both AT3ova and 4T1Ch9 tumor cells in vivo¹⁵. To further characterise the effect of MEKi on tumor immunogenicity we examined the expression of other receptors and ligands. We observed that the expression of Fas,

TRAIL and NKG2D (RAE-1) were significantly upregulated in the presence of MEKi in vitro in both the AT3ova and 4T1Ch9 cell lines (Figure 2 A, B). We also observed a trend for increased expression of FAS, TRAIL and NKG2D on AT-3ova tumor cells in vivo (Supplementary Figure 1A). Given that the most pronounced effects of MEKi were on MHC-I expression (Figure 2 A, B), we next explored the effects of MEKi induced MHC-I expression on tumor cells and subsequent T cell responses.”

4) In Figure 2, it is difficult to determine the effects of the treatment on T cell frequencies and/or function given how the data is presented. Total frequencies (% and total number) would be helpful, rather than fold-changes in expression.

Author’s response:

These points have been addressed in the responses to Reviewer 1 (**Question 6**). Please refer to Supplementary Figure 3 and Supplementary Figure 6 for frequencies displayed in FACS plots.

5) The proliferation data present in Figure 3 are intriguing, however it is not clear whether the addition of immune agonists has activity in this assay as there is little or no enhancement in proliferation with these agents alone. Are the cells somehow protected from the proliferative block in the presence of these other signals?

Author’s response:

It is clear that the immune agonists are active in this assay as shown by their ability to enhance the proliferation of MEKi treated T cells. We hypothesize that the reason there is no enhanced proliferation following the addition of the agonist antibodies in the absence of MEKi is because the T cells are already maximally stimulated with α -CD3/ α -CD28. However, in the context of MEKi the immune agonists are able to restore activation of the T cells through the p38/JNK pathways as shown in the analysis of T cell signaling in Figure 10. The importance of the JNK/p38 pathways in restoring T cell function is shown in Figure 10D where we show that the immune agonists are unable to rescue MEKi induced inhibition of IFN γ production in the context of either JNK or p38 inhibition.

6) It would be useful to evaluate the effects of MEKi plus aOX-40 or a4-1BB therapy in an in vivo priming model using OT-I T cells to elucidate the effects of this combination in a setting where a robust co-stimulatory signal is provided.

Author’s response:

We agree that this would be an interesting series of experiments. However, we feel it would constitute a whole new study, requiring several aspects of optimisation.

7) The data presented in Figure 4B is perplexing as there is no difference in IFN-g levels at 240 min post-treatment suggesting a very transient effect with MEKi. It isn’t clear why there isn’t a sustained effect on the MEKi monotherapy group. This should be discussed.

Author’s response:

These questions have been addressed in response to the comments of Reviewer 1 (**Question 7**).

8) The authors should discuss the tumor growth/survival data in Figure 5 – it is intriguing that none of the treatments led to complete tumor regression. Is this related to the MEKi treatment or just these particular tumor models? Have any other models been tested to see if durable CRs can be achieved?

Author’s response:

Reviewer 2 has made a similar point. Please see response to **Question 3**.

9) Also, recent work from the Ribas group (Oncoimmunology 5:7, e1052212; July 2016) should be discussed as they also examined the effects of BRAF inhibitors plus a4-1BB mAb therapy.

Author's response:

To address this, we have inserted the following text to the Discussion located on page 15.

"These results are similar to the findings of Moreno et al., where the authors showed that a quadruplet combination of a BRAFi, MEKi, either α -PD-1 or α -PDL-1 and either α -4-1BB or α -OX-40 significantly delayed tumor growth in BRAF mutant melanoma models³⁸. However, the effects of these combinations on survival were not shown."

10) Representative histograms and/or dot plots should be provided for the associated flow cytometry data presented in Figures 2 and 6.

Author's response:

This data has been included. Please refer to Supplementary Figure 3 and 6, referred to on page 6 and 9 of the Results section of the revised manuscript.

Reviewers' comments:

Reviewer #1 (Remarks to the Author):

The authors have addressed most of my prior questions, and the work is overall improved. However, the interpretation of some of the new data requires further consideration.

1) The authors state in lines 110-111 that "the most pronounced effects of MEKi were on MHC-I expression (Figure 2 A, B)". However, there is no significant MHC-I upregulation by trematinib in vitro without the addition of IFN γ . This point should be clearly mentioned in the manuscript because the dependency on IFN γ -producing T cells for the increased cancer cell immunogenicity induced by trematinib implies that when T cells are impaired in this function (i.e., in the absence of stimulatory antibodies to 41BB or OX40) trematinib will not increase tumor immunogenicity. Interestingly, data previously published by the same investigators (Clin Cancer Res 2016) show an increase in MHC-I by trematinib alone in vivo, suggesting that there is a source of IFN γ in vivo even in the absence of costimulatory antibodies, which may come from T cells or perhaps from innate immune cells. PDL-1 behaves like MHC-I, showing no induction by trematinib alone in vitro but an increase in vivo (based on data published in Clin Cancer Res 2016). However, there is no benefit of anti-PD-1 with trematinib on tumor growth or mice survival in the absence of either anti-41BB or anti-OX40 (Fig 7A-D), somewhat contradictory to what the authors previously published. These discrepancies should be addressed.

Of the other molecules tested in vivo, only Fas shows a trend to increased expression in trematinib-treated mice. However, only AT3 cells are shown (Supplementary Figure 1). Thus, the data are not fully supportive of a same trend in vivo, and this statement should be revised.

2) Trematinib by itself has a highly significant effect on DR5 and Fas in 4T1 cells in vitro (Figure 1B), but these cells are not analyzed in vivo. Fas upregulation could be as relevant as the upregulation of MHC-I for the increased anti-tumor effect when T cell function is rescued by co-stimulatory antibodies. T cells also kill via Fas and there are other examples when upregulation of Fas by treatment was shown to improve T cell-mediated tumor rejection (e.g., Chakraborty et al., J Immunol 2003). Likewise, NKG2D ligands can contribute to killing of targets by CD8 T cells. Thus, the statement that "the most pronounced effects of MEKi were on MHC-I expression and ... we next explored the effects of MEKi induced MHC-I expression on tumor cells and subsequent T cell responses" is not warranted since there are no experiments to unequivocally address the contribution of MHC-I versus the contribution of other upregulated receptors on T cell responses.

3) Experiments in RAG-deficient mice show that anti-41BB and anti-OX40 improve tumor control (Figure 9). No p values are shown but the effect seems to be significant. This raises the question if these antibodies are acting on innate immune cells stimulating their anti-tumor activities. Trematinib is very effective by itself in RAG $^{-/-}$ mice, and addition of the antibodies has no further effect, supporting the authors conclusions that the combination requires T cells. However, it is equally possible that trematinib itself acts on innate immune cells nullifying the effects of anti-41BB and anti-OX40. This alternative explanation should be considered and addressed experimentally.

Reviewer #3 (Remarks to the Author):

The revised version sufficiently addresses my concerns.

Response to reviewers' comments:

Reviewer #1

1) The authors state in lines 110-111 that “the most pronounced effects of MEKi were on MHC-I expression (Figure 2 A, B)”. However, there is no significant MHC-I upregulation by trametinib *in vitro* without the addition of IFN γ . This point should be clearly mentioned in the manuscript...

The reviewer is correct in their assertion that the effect of MEKi are more pronounced in the context of IFN γ but it is incorrect to state that MEKi has no significant effect on MHC-I expression by itself. As shown in Figure 3A, MEKi alone significantly ($P < 0.01$) upregulates MHC-I expression, due to cell intrinsic mechanisms. In Figure 2, the apparent lack of significance with MEKi alone is a consequence of the enormous difference following the addition of IFN γ , that is not accounted for by the one-way ANOVA test comparing multiple groups. The difference in the effects between vehicle and trametinib on MHC-I expression on both AT3ova and 4T1Ch9 tumor cells is significant when the other IFN γ groups are removed and a simple unpaired one-tailed student's t-test is performed. Please refer to the review only figure below. Therefore, the data indicates that MEKi can promote tumor cell immunogenicity by itself, but this is more pronounced in the presence of IFN γ which justifies our combination approach. To clarify this point, we have inserted the following text to the discussion on **page 13 line 316-325**:

“Interestingly, the effects of MEKi on MHC-I expression were most pronounced in the context of IFN γ . Whilst the molecular targets of MEKi leading to increased MHC-I expression remain undefined, this suggests that it may interact with signalling downstream of the IFN γ receptor. Nevertheless, this data clearly supports the concept that combining MEKi with immunotherapy, enhances IFN γ production, which has the potential to significantly enhance MHC-I expression and consequently tumor cell immunogenicity.”

Review only Figure:

2) PDL-1 behaves like MHC-I, showing no induction by trametinib alone *in vitro* but an increase *in vivo* (based on data published in Clin Cancer Res 2016). However, there is no benefit of anti-PD-1 with trametinib on tumor growth or mice survival in the absence of either anti-41BB or anti-OX40 (Fig 7A-D), somewhat contradictory to what the authors previously published. These discrepancies should be addressed.

Our experimental data for the MEKi and PD-1 combination therapy in the current study fully supports the data in the CCR paper. The AT3ova cell line was employed in the CCR paper for the combined

MEKi and PD-1 therapy whereas the data referred to by the reviewer in the current study is in the 4T1Ch9 model. The 4T1Ch9 cell line represents a low TILs/poorly -immunogenic and highly aggressive model, and unlike the AT3ova, it is known to be unresponsive to anti-PD-1 checkpoint blockade. This has been previously published by our group and others ^{1, 2, 3, 4}.

For clarification, there was no data shown in the CCR paper using this combination approach for 4T1Ch9 cells. Thus, our data suggest that the use of the immune agonists would appear to be a better option than anti-PD-1 for combination approaches with MEKi particularly for less immunogenic (cold) TNBCs. However, it should be noted that the best therapeutic effects observed in terms of long term survival of mice in both tumor models were following the triple combination of MEKi, anti-PD-1 and either anti-4-1BB or anti-OX-40 (Figure 7). To clarify this point we have added a statement regarding this in the discussion section on **page 13, lines 334-335** of the revised manuscript. Our findings therefore have immediate implications for translation into the clinical setting.

“Intriguingly, the triple combination had significant survival benefits in the low TILs model (4T1Ch9), which was resistant to the MEKi/α-PD-1 double combination therapy.”

3) Of the other molecules tested in vivo, only Fas shows a trend to increased expression in trematinib-treated mice. However, only AT3 cells are shown (Supplementary Figure 1). Thus, the data are not fully supportive of a same trend in vivo, and this statement should be revised.

We have addressed this concern in the Results section on **page 5, lines 106-112** of the revised manuscript.

“We observed that the expression of Fas, TRAIL and NKG2D (RAE-1) were significantly upregulated in the presence of MEKi in vitro in both the AT3ova and 4T1Ch9 cell lines (Figure 2 A, B). However, there were no significant changes in the expression of Fas, TRAIL and NKG2D on AT3ova tumor cells following MEKi treatment in vivo (Supplementary Figure 1A). Given that pronounced effects of MEKi were seen on MHC-I expression (Figure 2 A, B), we next explored the effects of MEKi induced MHC-I expression on tumor cells and subsequent T cell responses.”

4) T cells also kill via Fas and there are other examples when upregulation of Fas by treatment was shown to improve T cell-mediated tumor rejection (e.g., Chakraborty et al., J Immunol 2003). Likewise, NKG2D ligands can contribute to killing of targets by CD8 T cells. Thus, the statement that “the most pronounced effects of MEKi were on MHC-I expression and ... we next explored the effects of MEKi induced MHC-I expression on tumor cells and subsequent T cell responses” is not warranted since there are no experiments to unequivocally address the contribution of MHC-I versus the contribution of other upregulated receptors on T cell responses.

We agree that our results do not exclude a possible role for FASL;FAS and NKG2D:NKG2DL interactions in the mechanism by which the combination of MEKi and immunotherapy enhances anti-tumor effects. To address this point we have inserted the following text in the Discussion section on **page 13, line 321-325** of the revised manuscript.

“Although we have focussed upon increased expression of MHC-I as a mechanism of increased tumor immunogenicity in this study, our findings of increased expression of other tumor ligands such as Fas, TRAIL and NKG2DL(RAE-1) in vitro, suggests that further investigation into the relevance of these pathways in the observed therapeutic effects of MEKi is warranted.”

5) Experiments in RAG-deficient mice show that anti-41BB and anti-OX40 improve tumor control (Figure 9). No p values are shown but the effect seems to be significant. This raises the question if these antibodies are acting on innate immune cells stimulating their anti-tumor activities. Trematinib is very effective by itself in RAG-/- mice, and addition of the antibodies has no further effect, supporting the authors conclusions that the combination requires T cells.

However, it is equally possible that trematinib itself acts on innate immune cells nullifying the effects of anti-41BB and anti-OX40. This alternative explanation should be considered and addressed experimentally.

We agree with the reviewer that the MEKi could potentially be detrimental to innate immune cells, though the major focus of the current study is on T lymphocytes given their association with favorable outcome in TNBC (reviewed by Savas et al, Nat Rev Clin Oncol 2016). Nevertheless, we have performed additional experiments to analyse the effects of MEKi on CD3⁻ NK1.1⁺ NK cells and CD3⁺ NK1.1⁺ NKT cells. These experiments revealed that at day 4 post treatment, the total frequency of NK cells and NKT cells was unaffected by MEKi treatment. Similarly, MEKi had no effect on NK cell maturation (the proportion of CD11b⁺ CD27⁻ NK cells) or their expression of Granzyme B⁺.

This new data is referred to in the Results section on **page 7, lines 143-146** of the revised manuscript.

“Analysis of innate immune subsets such as NK cells and NK T cells revealed no changes in frequency (CD3⁺ NK1.1⁺, CD3⁻ NK1.1⁺), maturation (CD11b⁺ CD27⁻) or effector function (Granzyme B⁺) following MEKi (Supplementary Figure 4). As such, the focus of subsequent experiments was on the effects of MEKi on T cells specifically.”

This new data supports our previous observations that the detrimental effect of MEKi on anti-tumor immunity is predominantly due to suppression of conventional T cells. However, the possibility that MEKi also modulates other innate immune cells cannot be discounted, however is beyond the scope of this paper. Due to the large number of innate lymphoid subsets it is impossible to fully address this experimentally, however the major conclusion of the current study is that agonistic antibodies can rescue conventional T cells from MEKi induced suppression. To further highlight this point, we have inserted the following text into the Discussion section on **page 14-15, lines 371-374**.

“Whilst our study conclusively shows that MEKi is detrimental to T cell effector functions, the effect of MEKi on innate immune cells remains relatively unknown. In future studies, it will be interesting to characterise the requirement of MEK signalling in other immune cells.

1. Beavis PA, *et al.* Adenosine Receptor 2A Blockade Increases the Efficacy of Anti-PD-1 through Enhanced Antitumor T-cell Responses. *Cancer immunology research* **3**, 506-517 (2015).
2. De Henau O, *et al.* Overcoming resistance to checkpoint blockade therapy by targeting PI3Ky in myeloid cells. *Nature* **539**, 443-447 (2016).
3. Gao L, *et al.* Enhanced Anti-Tumor Efficacy through a Combination of Integrin $\alpha\beta 6$ -Targeted Photodynamic Therapy and Immune Checkpoint Inhibition. *Theranostics* **6**, 627-637 (2016).
4. Mall C, *et al.* Repeated PD-1/PD-L1 monoclonal antibody administration induces fatal xenogeneic hypersensitivity reactions in a murine model of breast cancer. *Oncoimmunology* **5**, e1075114 (2016).

REVIEWERS' COMMENTS:

Reviewer #1 (Remarks to the Author):

The authors have answered satisfactorily all of the questions concerning the interpretation of their results and made needed revisions to the manuscript. The results are exciting as they provide information that can guide a rationale clinical testing of combinations of targeted agents and immunotherapy.